# A verifiably secure and robust authentication protocol for synergistically-assisted IoD deployment drones

**Ali Delham Algarni**[1]*, **Nisreen Innab**[2], **Fahad Algarni**[1]

1 Faculty of Computing and Information Technology, University of Bisha, Bisha, Saudi Arabia, 2 Department of Computer Science and Information Systems, College of Applied Sciences, AlMaarefa University, Riyadh, Saudi Arabia

☉ These authors contributed equally to this work.

* al.alqarni@ub.edu.sa

**Data Availability Statement:** All relevant data are within the paper and its Supporting information files.

**Funding:** This study was financially supported by the Deanship of Scientific Research, University of

## Abstract

Drones have limited computing and storage resources, which makes them unable to perform complex tactical tasks; drones working in clusters can be employed for such complex tactical task completion. But, the synergy between these drones working in clusters is mandatory. Otherwise, the adversary can easily target them, disturb their routine work, and even disturb the whole system. Only a robust and lightweight authentication protocol can handle both information fusion and collaboration and coordination of drones working in clusters efficiently and effectively and is one of the essential components of the Internet of Drones (IoD) environment as it is deployed for quick and intelligent decisions. In this regard, the already available security mechanisms for IoD deployment drones have either design flaws or failed to provide security against session key disclosure, spoofing, man in the middle (MITM), and denial of service (DoS) attacks. Therefore, this article presents a protocol based on elliptic curve cryptography (ECC), one-way hash, concatenation, and exclusive-OR (XOR) operations to establish a secure communication session among drones which makes them a powerful system to perform complex tactical tasks and information fusion intelligently. The security analysis of the proposed protocol has formally been scrutinized via BAN logic and ProVerif and informally via realistic discussion and illustrations. The results obtained from the analysis sections demonstrate that the proposed protocol is robust in security and is 63.87% more efficient in terms of communication cost compared to its competitors, and 66.46% better in terms of computational cost.

## Introduction

The drone, an impressive technological innovation, operates without a human on board and can be controlled remotely through flying ad hoc networks (FANET), unmanned aerial vehicular networks (UAVN), etc. It is also compatible with GPS (Global Positioning System), 5G (Fifth Generation), and other wireless networking. Drones have many application areas,

Bisha, Saudi Arabia [https://www.ub.edu.sa] in the form of a grant (UB-Promising-26-1445). No additional external funding was received for this study.

**Competing interests:** The authors have declared that no competing interests exist.

including traffic monitoring, search/rescue operations, pipeline inspection, illegal trawling, wildlife surveillance, drug trafficking, smuggling, immigration, etc., which significantly enhance conventional surveillance methods into an intelligent way for information fusion and complex tactical task completion. However, the communication environment for drones, including with the ground control station (GCS), is hostile, with adversaries posing a threat and using it for malicious purposes [1]. Ensuring secure transmission between the drones and the GCS is a significant challenge, and many attempts have been made to protect the sensitive information exchanged in the IoD environment. The urgency and importance of security issues are yet to be solved, as without a robust security mechanism, the drone communication path remains vulnerable to strong adversaries, potentially leading to serious security breaches [2]. It is crucial to be aware of these risks and to work towards mitigating them.

Owing to limited computing and storage resources, a single drone cannot efficiently perform a complex tactical task; however, working in the cluster is qualified for complex operations subject to collaboration and coordination (synergy) as it is a crucial characteristic of the clustering drones when deployed for an intelligent information fusion. Due to this characteristic, the drones working in clusters can transmit information in a controlled manner to each other and send intelligent commands to the ground control station (GCS) for prompt decisions. The GCS plays a crucial and irreplaceable role in this process, as it is the central hub for receiving and processing data from the drones [3]. These synergistic characteristics require a flawless authentication mechanism to be installed in all drones working in clusters to make all the associated participants authenticated securely with each other and with the ground control station, detect suspicious drones in the sky, collect rich, real-time intelligence information related to the task for which the drones have been assigned, and periodically deliver these data to the GCS for information fusion. The potential risks of not having secure authentication are significant, as any dubious information from any drone may mislead the GCS to make a wrong decision. This underscores the need for immediate action to implement secure and robust authentication. There are two types of authentication: messaging and identification. The first is challenging, whereas the second is beyond the scope of this work because information authentication can ensure integrity, availability, confidentiality, nonrepudiation, etc. [4].

Numerous researchers have presented information authentications from time to time. For example, researchers [5] proposed an ECC-based lightweight authentication protocol for drone deployment in smart city surveillance. This protocol is significant as it utilizes discrete logarithmic problems (DLP) [6] for secure key generation, a crucial aspect in securing information communication in IoD technology. Another ECC-based authentication protocol for space information networks was presented by [7], who claimed that their protocol is secure and efficiently offers services between satellite and ground stations. They [7] analyzed the security of their framework through a well-known random oracle model (ROM), a theoretical model used in the analysis of cryptographic algorithms and protocols [8]. They [7] simulated it via the automated validation of the Internet Security Protocols and Applications (AVISPA) software toolkit [9]. A simple hash cryptographic-based lightweight and robust authentication scheme for the IoD environment was presented by [10], who scrutinized security via the Turing machine, lemmas, theorems, and informal discussion [11]. However, their [10] proposal lacks a password update facility and a drone revocation/reregistration/reissue phase. A service and temporal credential-based protocol for IoD deployment drones working in clusters was presented by [12], who claimed that their scheme is the first protocol to provide security for drone-collected data and is resistant to information leakage vulnerability. They used another well-known real-or-random (ROR) model [13], a model used to analyze the security of cryptographic protocols for security analysis, and the AVISPA tool for simulation [9]. They [12] claimed that their scenario is 20% smaller in computation than its competitors, indicating

potential efficiency gains. The three-factor key agreement protocol based on Boyko-Peinado-Venkatesan (BPV) was designed by [14], who claimed that the FourQ curve [15] is five times faster, lighter, and more robust in terms of security than other ECC conventional cryptographic primitives are. However, these schemes either have design flaws or fail to provide security features such as authentication, confidentiality, and privacy. Additionally, these schemes are not secure against session key disclosure attacks, spoofing, man in the middle, denial of service, etc., attacks.

Similarly, the researchers in [16] demonstrated their dedication to the field by arguing that desynchronization, MITM, replay, tracing, and side-channel attacks must be addressed efficiently in IoD environments. They [16] proposed a physical unclonable function (PUF)-based [17] authentication protocol to mitigate these drawbacks. Their [16] commitment to finding solutions was evident in their claim that the security analysis of their strategy was not just comprehensive but flawless and that their simulation results were more substantial than those in other studies. They [16] also used a hybrid cryptosystem to mitigate critical vulnerabilities, a solution they claimed was more effective than existing schemes but with a more complex practical implementation. A blockchain-based security scheme was introduced in [18], and they claimed that their scheme provides secure data transmission in a 5G-enabled IoD system, with advantages over existing schemes. Their [18] blockchain-oriented data-delivery collection security authenticated all the participants of IoD and resisted numerous threats, whereas a buffer pseudonym and public key infrastructure (PKI)-based authentication scheme for an edge-assisted IoD system was designed in [19]; however, their computational costs are too high compared with those of existing schemes.

The researchers in [20] also introduced a communication-aware drone delivery problem (C-DDP) for IoD to ensure efficient last-mile delivery operations. Their [20] work enhances the trade-off among the drones working in the clusters and minimizes unintentional interference in shared spectrum scenarios. Their [20] study highlighted the impact of UAV interference on ground users, emphasizing the urgency and significance of their proposed solutions. Another study [21] stressed the importance of optimizing communication strategies, with a particular focus on factors such as the signal-to-noise ratio, outage duration thresholds, and interference tolerance levels, while [22] added a layer of complexity and depth to the flying zone by enhancing drone-to-drone communication reliability and efficiency. The complexity of the flying zone necessitates advanced communication strategies. Achieving such reliable communications in UAV networks is challenging because of the open networking path and the requirement of flawless and robust authentication, which is often missing and, in turn, can cause unintentional interference to the ground control station (GCS) [23]. However, these methods [23–25], have some drawbacks because when drones are flying, they can affect people's privacy and create a power optimization issue due to the shared spectrum, and communication links usually fail in the route traffic as drones fly at high speed. Therefore, these issues can be solved only by designing a synergistically-assisted protocol with efficient information fusion in the IoD environment. The proposed solutions are of utmost importance and urgent in the complex and dynamic environment of drone technology.

## Objectives

This research work aim to ensure secure communication between two drones in the IoD environment, in this regard, the objectives of this research work includes:

1. Drones often communicate with other drones, GCSs, or other devices through wireless networks. Secure communication is crucial to prevent unauthorized access or tampering. A

protocol can be designed to compute a shared session key among drones, ensuring that keys are securely exchanged and authenticated.

2. The proposed protocol is actually an implicit key authentication method in which the drone technology should be offered a strong layer of security by ensuring that the exchange of secrets among drones are authenticated without additional steps, thereby preventing unauthorized drones from spoofing or hijacking the communication channel.

3. The proposed protocol is designed with a strong focus on resistance to attacks. It must have the ability to provide security properties that ensure resistance against various attacks, such as eavesdropping, message tampering, and impersonation.

4. The proposed authentication protocol is specifically designed to ensure that the session keys used for communication among drones remain secret. This is not just a security measure but a guarantee that data interception and manipulation are prevented, thereby ensuring the integrity and confidentiality of the communication process.

5. The proposed authentication protocol must have a formal framework for analyzing the security of communication protocols. A thorough analysis, which involves applying BAN logic, a formal method for analyzing the security of cryptographic protocols, is crucial in identifying and addressing vulnerabilities before deployment when dealing with drone communication protocols.

## Motivation and contributions

Many vulnerabilities, issues, challenges, and limitations have been noted in existing authentication methods, as discussed in detail in the above protocols. To address these issues, an innovative approach, such as using random numbers, identities, and secret keys, enhances and mitigates the identified security lapses, flaws, and security issues. Additionally, by leveraging the unique characteristics of drones, such as their mobility, agility, and ability to operate in remote areas and mitigate known vulnerabilities, issues, and challenges, authentication presents a promising solution that is resilient to attacks and robust under diverse conditions. The robustness of such a solution provides reassurance about the security of the IoD environment. By incorporating these advancements, the development of an effective authentication protocol can significantly increase the overall security and reliability of the IoD environment because drones transmit information via an open channel that is vulnerable to various attacks, such as impersonation, password guessing, insider attacks, denial of service, and replay attacks. Rigorously, the authentication of drones working in clusters is crucial to prevent critical security breaches and build trust in the system. Authentication also measures challenges like key secrecy, heavyweight computations, high communication costs, and dependence on random numbers, so the major contributions of this research work are as follows:

- To present a protocol that provides mutual authentication and efficient services in an IoD environment, a secure session key should be established among the drones working in the cluster for secure communication without interacting with the ground control station.

- To utilize the elliptic curve cryptography (ECC), a one-way hash function, and XOR operations for the design of the proposed protocol in which the GCS securely stores shared credentials in the memory of each drone at the time of registration. Once deployed for tactical tasks, the drones exchange these credentials, enabling secure authentication and start communication in a collaborative and coordinated manner.

- To formally analyze the proposed verifiably secure and robust authentication protocol through BAN logic, a widely accepted method for evaluating cryptographic protocols, and ProVerif simulation, a tool for verifying the security properties of cryptographic protocols.

- To conduct a comparative analysis of the proposed synergistically-assisted IoD deployment drone protocol regarding performance metrics and security functionalities. This analysis confirms that the protocol is lightweight, robust, and synergistically assisted for the IoD environments, highlighting its practical application.

### Paper layout

The remainder of the paper is organized as follows: In the preliminaries and backgrounds section of the article, we have presented the groundwork pertaining to conducting this research work. In the literature review section, we have presented the related work comprehensively, including methodologies and pros and cons of their work, and analyzed the baseline scheme for vulnerabilities like key disclosure, forward secrecy and stolen-verifier attacks. In the proposed protocol section, we have presented a practical and efficient protocol for an IoD environment where two drones can successfully share secrets and communicate securely. Afterwards, we analyzed the proposed protocol, formally using BAN logic and ProVerif validation and informally through pragmatic discussion. Then, we measured the performance metrics by considering communication, computational, and storage costs. This analysis shows that the proposed protocol is superior to all, and the work concludes with the optimistic note that the proposed protocol is ready for practical implementation in the IoD environment, offering hope for the future of secure drone communication.

## Preliminaries and background

This section provides the essential context, foundational information, and relevant background knowledge necessary for understanding the research's main content. It serves to orient the reader, provides the essential context, and sets the stage for the article's main content.

### Elliptic Curve Cryptography (ECC)

ECC, a methodology introduced by in 1985 [26], is elegantly defined through the simple equation $y^2 = x^3 + ax + b$. The private key in ECC is an integer; when multiplied with any fixed point of the curve, we obtain another key P called a public key. This process, along with the security features of ECC, ensures that any two points on the curve, $P(x_1, y_1)$ and $Q(x_2, y_2)$, if they intersect, generate a third point $R(x_3, y_3)$ on the curve, which should be equivalent to the addition of P and Q. Therefore, we use this technique for designing the proposed protocol to exchange data from one drone to another securely, providing reassurance about the safety of the process.

### Hash function

The National Security Agency (NSA) devised the Secure Hash Algorithm (SHA), a versatile tool in the field of cryptography. This algorithm, which was later released in 1993 by the esteemed National Institute of Standards and Technology (NIST), can handle input data of any size and generate a 160-bit (20-byte) hash value known as a message digest, usually shown as a hexadecimal number [27].

## XOR operations

In cryptography, the bitwise XOR (exclusive OR) is frequently utilized in security applications for several functions, such as data integrity checking, checksum computation, and encryption, also called a "*One-Time Pad*." XOR operations' straightforwardness, effectiveness, and bitwise ciphering nature make it a key component in various security-related applications. However, it is important to remember that they are not used as stand-alone security measures but rather as components of more complex cryptographic algorithms [28].

## Security requirements

To ensure secure communication between two drones in the IoD environment, the following security requirements are implemented:

**1-Key Exchange Protocol**: Drones often communicate with other drones, GCSs, or other devices through wireless networks. Secure communication is crucial to prevent unauthorized access or tampering. The proposed authentication protocol can be used to compute a shared session key among drones, ensuring that keys are securely exchanged and authenticated.

**2-Implicit Key Authentication** is a key feature in drone technology that provides a strong layer of security. It ensures that the keys exchanged between the first and second drones are authenticated without additional steps, thereby preventing unauthorized drones from spoofing or hijacking the communication channel. This measure provides a sense of reassurance about the security of the communication channel.

**3-The proposed protocol is** designed with a strong focus on resistance to attacks. It must have the ability to provide security properties that ensure resistance against various attacks, such as eavesdropping, message tampering, and impersonation. By designing drone communication protocols based on the CK model [30], a widely accepted model for cryptographic key exchange, vulnerabilities to these attacks can be effectively mitigated. This emphasis on security measures instills confidence in the audience.

**4-Session Key Secrecy**: The proposed authentication protocol is specifically designed to ensure that the session keys used for communication among drones remain secret. This is not just a security measure but a guarantee that data interception and manipulation are prevented, thereby ensuring the integrity and confidentiality of the communication process.

**5-Formal Analysis**: The proposed authentication protocol must have a formal framework for analyzing the security of communication protocols. A thorough analysis, which involves applying BAN logic, a formal method for analyzing the security of cryptographic protocols, is crucial in identifying and addressing vulnerabilities before deployment when dealing with drone communication protocols.

## Threat model

The Dolev-Yao [29] and Canetti-Krawczyk [30] models, which were the first to present the possible threats to the system, are known for their thoroughness in the field of cyber security. These models, widely used for protocol analysis and detection of system vulnerabilities, are the foundation of this research work. Their comprehensive nature in identifying threats and understanding potential weaknesses or loopholes provides a robust framework for our

research. According to these models, the system faces several threats, some of these threats includes the following:

- **False Data Injection Threat:** The adversary's ability to influence the information gathered by the drones, including state variables, coordinates, and preferences, falsifies valuable credentials and poses a significant risk to the system's integrity and operation.

- **Privacy Threat:** The adversary might access the open network channel, installing aircrack-ng [31] to crack the Wi-Fi password, airodump-ng [31] to capture data packets, and airplay-ng [31] to disconnect devices from the network, thereby compromising the privacy of a legitimate drone.

- **Traffic Analysis Threat:** The packets transmitted among two drones can be captured by an adversary, analyzed, and later used for malicious acts. The drones are equipped with sensors/cameras for data collection and exchange with the GCS alongside another drone. The transmission is performed via wireless media and is vulnerable to the adversary. If the security mechanism is weak, the adversary can easily extract the internal secret credentials from the exchanged packets.

- **Access Control Threat:** The adversary's ability to identify and exploit the different policies, rules, and design principles about a protocol's design can lead to a loss of authenticity, unauthorized privilege changes, and significant system damage. This underscores the urgent need for robust access control measures.

- **Identity Spoofing Threat:** The adversary's ability to masquerade as someone else to gain unauthorized access to sensitive information, commit fraud, or carry out other illicit activities highlights the need for vigilance and proactive measures to prevent such threats.

- **Reply Attack:** An attacker captures data from the open channel, eavesdrops, and later uses it for a potential replay attack to gain access to the system illegally.

- **Desynchronization Threat:** A desynchronization threat occurs when a system administrator updates a legitimate drone's identity and other credentials; however, the drone does not know about these changes. Conversely, an adversary can access the GCS and disturb the synchrony of the shared memory for deployed drones.

- **Man-in-the-middle (MITM) attack:** The adversary places itself between two communicating parties and relays messages for them instead of transmitting them to a legitimate drone or GCS. The attacker can then divert and modify the messages' contents and impersonate the whole system.

- **Stolen-Verifier Attack**: The attacker's ability to steal a drone, extract its internal secret credentials, and launch various attacks on the system, including replay, masquerade, and denial-of-service (DoS) attacks, can severely damage the system's security and functionality. This underscores the need for comprehensive security measures.

Similarly, the Canetti and Krawczyk [30] model provides a reliable framework for analyzing the security of cryptographic protocols. It defines a security notion called "implicit key authentication," which captures the requirement that the shared secret key should be authenticated implicitly during the key exchange process. In this process, two parties agree on a shared secret key without additional authentication steps. This model is useful for setting rules for a cryptographic protocol and is considered to be secure if it satisfies specific security properties defined by [30], such as key secrecy, key authentication, forward, backward, and session key confidentiality. These properties ensure that the protocol resists various attacks, such as

impersonation attacks, false data injection attacks, privacy issues, traffic analysis attacks, access control threats, identity spoofing vulnerabilities, reply attacks, desynchronization issues, man-in-the-middle (MITM) attacks, stolen-verifier attacks, and key compromise attacks.

## System model

The proposed system or network model, a crucial advancement in drone technology [32] and telecommunications [33], consists of two main entities, a drone and a GCS, diagrammatically represented in Fig 1.

The entities showing in Fig 1 are drone and ground control station (GCS); these are define one by one as under:

**Drone (D):** The operator's remote supervision of drones is a necessary aspect of this system, but it's not just a one-way process. Even if there are no onboard operators to respond to emergencies, overseeing tasks such as farming, wildlife, surveillance, and labor monitoring for work output in a big building requires continuous investigation. These drones are not autonomous entities but rather tools that require human collaboration to improve mission efficacy while minimizing costs. This article proposes mutual authentication to establish secure communication between two drones in the airspace, emphasizing the collaborative nature of the system. The two drones can incorporate global positioning system (GPS) signals alongside additional wireless communication layouts to communicate with the flying ad hoc network (FANET) or unmanned aerial vehicular network (UAVN) or use 5G/6G.

**Ground Control Station (GCS):** The GCS, a pivotal component in the system, is a specialist service offering connections, assistance with data analysis, and real-time capabilities for

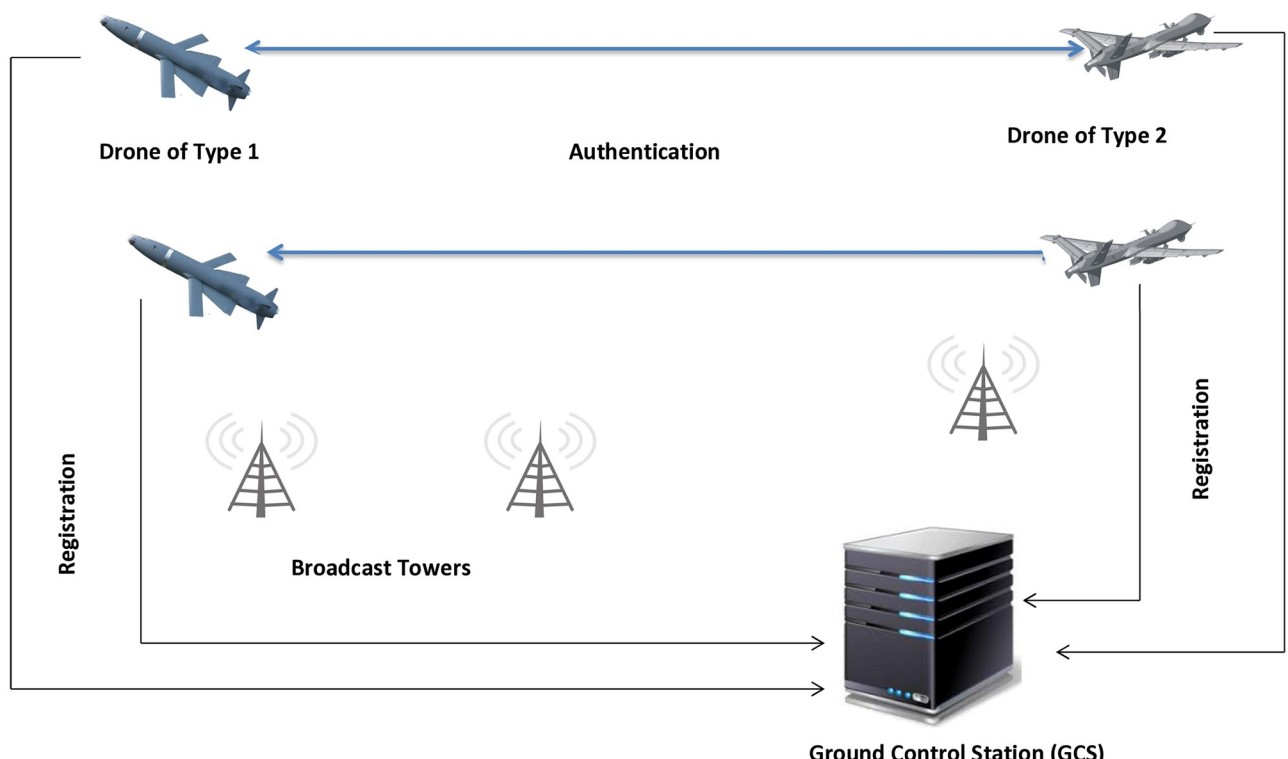

**Fig 1. Proposed system model.**

handling problems. It plays a key role in managing, overseeing, and regulating drones to provide navigational services. Every drone must have the GCS installed and be linked with other network services, such as 5G/6G, GPS, UAVN/FANET, and wireless communication gateway. Installing drones inside a designated flying zone and establishing them together in pre-established flight zones is mandatory. The GCS controls the drone's flight and verifies its identity within the zone. The GCS also makes it simple to identify illegitimate drones in the flying area or verify the legitimacy of a valid drone.

## Related works

This section offers a brief overview and evaluation of the previous state-of-the-art works. It assesses the advantages, disadvantages, and overall contributions of different schemes presented by other researchers in the area in the form of a table. It also offers a thorough and critical summary of the existing body of research, highlighting the urgency of identifying gaps, contradictions, and areas needing further investigation. The summary of the related works is shown in Table 1, underscoring the need for future research to address these gaps.

In summary, research on drone authentication techniques is highly important. It discusses the use of three-factor user authentication and elliptic curve cryptography (ECC), SHA, and XOR operations to enhance resource-efficient security for IoDs. However, these techniques are not without vulnerabilities, such as susceptibility to impersonation, replay, denial of service (DoS), parallel session, and secret credential leakage attacks. This underscores the need for robust security measures in drone technology.

### Review analysis of the baseline scheme

Jan et al. [33] proposed a biometric fuzzy extractor-based scheme for the cross-verification of users and drones in the IoD environment. Their scheme has two main phases, registration and verification, which are explained in the following subsection. The notations used for the design of their scheme are shown in Table 2.

**Registration phase.** The registration phase in [33] is completed in the following steps:

1. This phase is completed by first registering a user with the GCS and then a drone. The identity and password are sent to the GCS through a secure channel for user registration.

2. When receiving the registration message from a user, the GCS calculates $PID_i = h(ID_i||s)$, $A_i = h(ID_i||l)$, where $l$ is a large secret number and $s$ is the public key. The GCS injects $\{PID_i, A_i, ID_i\}$ in its memory and replays $\{PID_i, A_i\}$ to the user.

3. The user then receives the response message from the GCS and is asked to enter their biometrics $BIO_i$, calculate $Gen(BIO_i) = (\sigma_i, \tau_i)$, $A_i = h(ID_i||PW_i)\oplus A_i$, and $PID_i = h(ID_i||PW_i)\oplus PID_i$ and inject $\{A_i, PID_i, Gen(.), Rep(.)\}$ in their mobile device memory.

4. The drone is also registered with the GCS through a secure channel and sends its identity to the GCS.

5. The GCS calculates $PID_j = h(ID_j||s)$ and $A_j = h(ID_j||l)$, injects $\{ID_j, PID_j, A_j\}$ and replays $\{PID_j, A_j\}$ back to the drone.

6. The drone also injects $\{PID_j, A_j\}$ into the device memory.

**Verification phase.** As detailed in [33], this phase is accomplished in the following steps:

**Table 1. Critical literature review summary.**

| Ref | Year | Methodology | Description | Advantages |
|---|---|---|---|---|
| [34] | 2020 | ECC | The researchers argued that energy limitations in UAV environments impact authentication suitability, restrict fly times, and lack a fast authentication process. | Their proposed framework ensures data confidentiality and mutual authentication, prevents potential attacks, and has lower communication costs than existing schemes. |
| [35] | 2020 | PUF | UAVs are used for various applications, but the security of UAV networks, along with fast service delivery, is a critical need. Hence, they used PUF and lightweight cryptographic methods to design an authentication protocol. | The researchers proposed a mutual authentication protocol for software-defined UAV networks. They successfully achieved mutual authentication across three levels of entities: user, drone, and GCS. They provided formal logic proof and comparisons. |
| [36] | 2021 | RFID | The researchers focused on RFID security, authentication schemes, and UAV applications. | Their proposed protocol is lightweight and provides robust security. |
| [37] | 2022 | PKI | The researchers focused on securing the internet by proposing a robust authentication protocol based on the PKI method. They highlighted deficits in prior protocols, offered an improved security solution, and formally analyzed security using BAN logic and ProVerif2.03 simulation. | Their proposed protocol is efficient and effective, generating secure keys among users, drones, and CSS. |
| [38] | 2022 | SHA-256 | The researchers proposed a resource-efficient authentication protocol for IIoT and named it REAP-IIoT. They employed a lightweight cryptographic technique to secure communication in the IIoT environment. Their scheme also ensures a user's privacy by securely authenticating and encrypting their personal sensitive information, and they validate their scheme against various security attacks. | The REAP-IIoT protocol ensures encrypted communication in IIoT applications and provides better security features with low overheads. The comparative analysis shows that REAP-IIoT requires less computational and communication costs. |
| [39] | 2023 | SHA-1 | The researchers focused on drone-based cloud architecture for secure data collection. They proposed a lightweight security protocol that ensures secure communication and mutual authentication. | They used the Scyther simulator to confirm that the protocol does not have security loopholes and that private information will not leak during execution. Their scheme is lightweight, efficient, and secure against insider attacks. |
| [40] | 2023 | Blom and Stream Cipher | The researchers proposed a scheme for enhancing aerial vehicle security via encryption and authentication, optimized their crucial distribution and updating process for efficient session key management, and utilized Grain stream cipher and MAC authentication for message security. Their security analysis confirmed the scheme's resistance to attacks and real-time performance. | Their proposed scheme ensured AV security with lightweight encryption and authentication, and their performance analysis shows that the scheme meets real-time requirements with minimal busload. |
| [41] | 2024 | PUF | The researchers proposed a secure lightweight authentication protocol for drone communication, which they named PRLAP-IoD. This protocol can resist various attacks, such as device loss and impersonation, by validating security through the ROR model, AVISPA tool, and informal analysis. | PRLAP-IoD provides robust security, defends against known attacks, effectively balances computation and communication costs in IoD, and surpasses existing IoD, AKA schemes in computation cost. |
| [42] | 2024 | Asymmetric Cryptography | The researchers proposed a novel authentication scheme for the internet of Drones to enhance security and performance. | Their proposed protocol outperformed its competitors in communication and computation and achieved a balance of security and performance missing in other schemes. |

PUF: physical unclonable function, RFID: radio frequency identification, SHA: secure hash algorithm, HMAC: hash message authentication code, MAC: message authentication code, AKA: authentication and key agreement, IIoT: industrial internet-of-Things, BAN: Burrows–Abadi–Needham, CSS: control system server, PKI: public key Infrastructure

**Table 2. Notations and their meaning.**

| Notation | Meaning | Notation | Meaning |
|---|---|---|---|
| $ID_i$ | User identity | $l$ | Secret number |
| $Gen(.)$ | Biometric Generation | $Rep(.)$ | Biometric Replication |
| $PW_i$ | User Password | $BIOi_i$ | User Biometrics |
| $s$ | Secret key | $ID_j$ | Drone identity |
| $R_1, R_2$ | Random numbers | $ST_1$ | System time |
| $SK$ | Session Key | $\|$ | Concatenation |
| $h(.)$ | Hash Function | $\oplus$ | XOR operations |

1. The user provides their identity, password, and biometrics. The device with the user calculates $\sigma_i^* = Rep(BIO_i, \tau_i)$, $PID_i = PID_i \oplus h(ID_i||PW_i)$, $A_i = A_i^* \oplus h(ID_i||PW_i)$, selects $R_1$, $ST_1$ and calculates $M_1 = h(PID_j||ST_1) \oplus PID_i$, $M_2 = h(PID_i||PID_j||A_i) \oplus R_1$, $M_3 = h(PID_i||PID_j||A_i||R_1) \oplus ID_i$, $M_4 = h(PID_i||PID_j||PID_k||A_i||R_1)$ and sends $Mesg_1 = \{M_1, M_2, M_3, M_4, ST_1\}$ to the GCS.

2. The GCS confirms the time validity, calculates $PID_i^* = M_1 \oplus h(PID_k||ST_1)$, extracts $A_i^*$ and computes $R_1^* = M_2 \oplus h(PID_i^*||PID_k||A_i^*)$, $PID_j^* = M_3 \oplus h(PID_i^*||PID_k||A_i^*||R_1^*)$ and $M_4^* = h(PID_i^*||PID_j^*||PID_k||A_i^*||R_1^*)$, validates $M_4^*? = M_4$, selects $ST_2$, calculates $M_5 = h(PID_j^*||A_j^*||ST_2) \oplus R_1^*$, $M_6 = h(PID_j^*||PID_k||A_j^*||R_1^*) \oplus PID_i^*$, $M_7 = h(PID_i^*||PID_j^*||PID_k||A_j^*||R_1^*)$ and sends $Mesg_2 = \{M_5, M_6, M_7, ST_2\}$ to the drone.

3. The drone first confirms the timestamp and calculates $R_1^{**} = M_5 \oplus h(PID_j||A_j)$, $PID_i^{**} = M_6 \oplus h(PID_j||PID_k||A_j||R_1^{**})$ and $M_7^* = h(PID_i^{**}||PID_j||PID_k||A_j||R_1^{**})$, validate $M_7^*? = M_7$, selects $R_2$, $ST_3$ calculates $M_8 = h(PID_j||PID_i^{**}||ST_3) \oplus R_2$, $M_9 = h(R_1^{**}||R_2)$, $SK_{ij} = h(PID_i^{**}||PID_j||PID_k||M_9)$, $M_{10} = h(PID_i^{**}||PID_j||PID_k||R_1^{**}||R_2||M_9)$, replays $Mesg_3 = \{M_8, M_9, M_{10}, ST_3\}$ back to the GCS.

4. The GCS confirms the timestamp, calculates $R_2^* = M_8 \oplus h(PID_j||PID_i||R_1)$, $M_9^* = h(R_1||R_2^*)$, and $M_{10} = h(PID_i||PID_j||PID_k||R_1||R_2^*)$, validates $M_{10}^*? = M_{10}$, calculates $SK_{ij} = h(PID_i||PID_j||PID_k||M_9^*)$, and replays $Mesg_4 = \{M_8, M_9, M_{10}, ST_4\}$ to the user.

5. The user confirms the timestamp, calculates $R_2^* = M_8 \oplus h(PID_j||PID_i||R_1)$, $M_9^* = h(R_1||R_2^*)$, $M_{10} = h(PID_i||PID_j||PID_k||R_1||R_2^*)$, verifies $M_{10}^*? = M_{10}$, calculates $SK_{ij} = h(PID_i||PID_j||PID_k||M_9^*)$ and keeps it as a shared session key.

## Cryptanalysis of the baseline scheme

After investigating the scheme presented by [33], the following loopholes/vulnerabilities/weaknesses are noted.

**Key disclosure attack.** The adversary becomes successful if they can obtain $PID_i$, $PID_j$, and $PID_k$. To do so, they first have to select two random numbers $R_{A1}$ and $R_{A2}$, pass through a hash function to obtain $M_9 = h(R_{A1}||R_{A2})$ and compute $M_5 \oplus h(PID_j||A_j)$, $M_5 = h(PID_i||A_j)$ and $PID_i = M_5 \oplus h(ID_j||l)$ to obtain $PID_i$. For $PID_j$, the adversary computes $M_8 \oplus h(PID_j||PID_i||R_1)$, $M_8 = h(PID_j||PID_i||R_1)$, and $PID_j = M_8 \oplus h(PID_i||R_1)$. Now, to obtain $PID_k$, the adversary computes $M_2 \oplus h(PID_i||PID_k||A_i)$, $M_2 = h(PID_i||PID_k||A_i)$, and $PID_k = M_2 \oplus h(PID_i||A_i)$. By obtaining $PID_i$, $PID_j$ and $PID_k$, the adversary can easily compute the session secret key $SK_{ij} = h(PID_i||PID_j||PID_k||M_9)$ in an easy manner. Therefore, the method protocol in [33] is vulnerable to session key disclosure attacks.

**Perfect forward security issue.** The adversary compromises all the previously transmitted messages $Mesg_1$, $Mesg_2$, $Mesg_3$, and $Mesg_4$ and reaches the secret keys $s$ and $l$. For example, a legitimate user enters their identity password, imprints biometrics and computes $\sigma_i^* = Rep(BIO_i, \tau_i)$, $PID_i = PID_i^a \oplus h(ID_i||PW_i)$, $A_i = A_{ia}^a \oplus h(ID_i||PW_i)$, selects $R_1$, $ST_1$ and calculates $M_1 = h(PID_j||ST_1) \oplus PID_i$, $M_2 = h(PID_i||PID_j||A_i) \oplus R_1$, $M_3 = h(PID_i||PID_j||A_i||R_1) \oplus ID_i$, $M_4 = h(PID_i||PID_j||PID_k||A_i||R_1)$ to construct $Mesg_1 = \{M_1, M_2, M_3, M_4, ST_1\}$. The adversary obtains $Mesg_1$, selects their own timestamp $T_A$, verifies $T_A - TS_1 \leq \Delta T$ and easily computes $M_3 = h(PID_i||PID_j||A_i||R_1) \oplus ID_i$ and $ID_1 = M_3 \oplus h(PID_i||PID_j||A_i||R_1)$. Then, the adversary takes two random numbers $R_{A1}$ and $R_{A2}$ and computes $M_9 = h(R_{A1}||R_{A2})$ and can easily compute $SK_{ij} = h(PID_i^{**}||PID_j||PID_k||M_9)$. Thus, the secrecy of the method in [33] is compromised, and it cannot deliver reliable services to the IoD environment.

**Table 3. Symbols and their meanings.**

| Symbols | Pronounced as | Symbols | Pronounced as |
|---|---|---|---|
| $D_1$ | First Drone | $P_{D1}$ | Public Key of the first drone |
| $D_2$ | Second Drone | $P_{D2}$ | Public Key of the second drone |
| GCS | Ground Control Station | $P_{GCS}$ | Public key of GCS |
| $ID_{D1}$ | Identity of the first drone | s | Private key of GCS |
| $ID_{D2}$ | Identity of the second drone | $d_1$ | Private key of the first drone |
| $L_{D1}$ | A random number of the first drone | $d_2$ | Private key of the second drone |
| $N_{D1}$ | None of the first drone | $KS_{DD}$ | Shared key |
| $L_{D2}$ | Random number of the second drone | $SID_{D1}$ | Pseudo-Identity of the first drone |
| $N_{D2}$ | Nonce of the second drone | $SID_{D2}$ | Pseudo-Identity of the second drone |
| $L_{GCS}$ | Random number of the GCS | II | Concatenation Function |
| T | Timestamp | $\oplus$ | XOR operations |

**Stolen-verifier attack.** Suppose that someone has stolen a mobile device from a user and wants to extract the user's sensitive internal credentials. In this case, they can easily obtain this information because it is available in $\{A_i^a, PID_{ia}^a, Gen(.), Rep(.)\}$ by passing $Gen(BIO_i) = (\sigma_i, \tau_i)$, $A_i^a = h(ID_i||PW_i) \oplus A_i$, and $PID_i^a = h(ID_i||PW_i) \oplus PID_a$ steps through the reverse engineering technique. Similarly, the drone memory has $\{ID_j, PID_j, A_j\}$ by computing $PID_j = h(ID_j||s)$ and $A_j = h(ID_j||l)$. Therefore, the method in [33] is not resistant to stolen-verifier attacks.

## Proposed protocol

The drone-to-drone communication security framework has three main phases: setup, registration, and mutual authentication, as described in the following subsection. The symbols used for designing the scheme are shown in Table 3.

### Setup phase

The GCS selects a point P from the elliptic curve $E(F_P)$ over $E_P(x, y)$, chooses a secret key s, and calculates the ECC-based public key $P_{GCS} = s. P$, selects h(.) and publishes public parameters $\{E(x, y), P, P_{GCS}, h(.)\}$ and secret key *s*.

### Registration phase

Every drone first registers with the GCS and then is deployed for tasks in the IoD environment. The registration is performed once in offline mode, and the real identity of each drone is communicated over a secure channel. The process of registration is undertaken in the following steps:

**Drone1 registration.** This phase of the protocol is accomplished in takes the following steps of computations:

**Step 1:** The first drone selects an identity $ID_{D1}$ and random number $L_{D1}$, calculates $RD_1 = h(ID_{D1}||L_{D1})$ and sends $\{ID_{D1}, RD_1\}$ to the GCS.

**Step 2:** The GCS, after receiving the $\{ID_{D1}, RD_1\}$ message, checks the identity of the drone in the record; if it is found, the GCS sends a message to the drone to select a unique identity; if not found, the GCS invokes its public key $P_{GCS}$ and computes $Y_{D1} = P_{GCS} \oplus h(ID_{D1}||L_{D1}||s)$.

**Step 3:** The GCS now calculates the pseudo-identity for the first drone, where $SID_{D1} = h(Y_{D1}. T||ID_{D1})$ and pseudo-identity for the second drone, $SID_{D2} = h(Y_{D2}. T||ID_{D2})$ and builds a

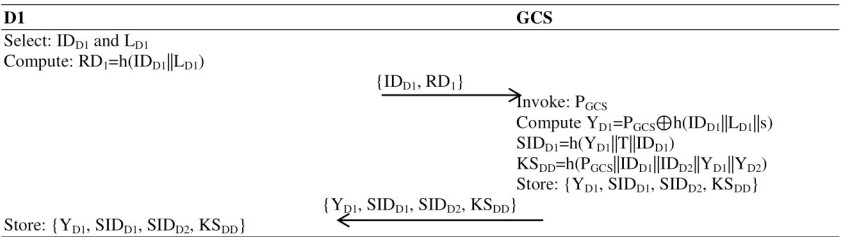

**Fig 2. Registration of the first drone (D1) with the GCS.**

long-term shared key $KS_{DD} = h(P_{GCS}||ID_{D1}||ID_{D2}||Y_{D1}||Y_{D2})$ and sends $\{Y_{D1}$, $SID_{D1}$, $SID_{D2}$, $KS_{DD}\}$ to the first drone and second drone. Now, the memory of both the first and second drones contains $\{Y_{D1}, SID_{D1}, SID_{D2}, KS_{DD}\}$ parameters, as shown in Fig 2.

**Drone2 registration.** This phase takes the following steps:

**Step 1:** The second drone selects an identity $ID_{D2}$ and random number $L_{D2}$, calculates $RD_2 = h(ID_{D2}||L_{D2})$ and sends $\{ID_{D2}, RD_2\}$ to the GCS.

**Step 2:** The GCS, after receiving the $\{ID_{D2}, RD_2\}$ message, checks the identity of the drone in the record; if it is found, the GCS sends a message to the drone to select a unique identity; if not found, the GCS invokes its public key $P_{GCS}$ and computes $Y_{D2} = P_{GCS} \oplus h(ID_{D2}||L_{D2}||s)$.

**Step 3:** The GCS now calculates the pseudo-identity for the first drone, where $SID_{D1} = h(Y_{D1}.$ $T||ID_{D1})$, and the pseudo-identity for the second drone is $SID_{D2} = h(Y_{D2}. T||ID_{D2})$ and builds a long-term shared key $KS_{DD} = h(P_{GCS}||ID_{D1}||ID_{D2}||Y_{D1}||Y_{D2})$ and sends $\{Y_{D1}$, $SID_{D1}$, $SID_{D2}$, $KS_{DD}\}$ to the first drone and second drone. Now, the memory of both the first and second drones contains $\{Y_{D1}, SID_{D1}, SID_{D2}, KS_{DD}\}$ parameters, as shown in Fig 3.

## Mutual authentication

The mutual authentication is undertaken in two round trips or three computation steps, explained as follows:

**Step 1:** The first drone selects timestamp T, computes $Z_1 = h(KS_{DD} \oplus P_{D1})$, recovers $SID_{D1}$ from storage, generates a nonce $N_{d1}$ and computes $Z_2 = N_{d1}. P$, $Z_3 = N_{d1}. P_{D2}$, $Z_4 = SID_{D1}|| Z_1||T) \oplus (Z_3||KS_{DD})$ and replays $\{Z_3, Z_4\}$ to the second drone over an insecure channel.

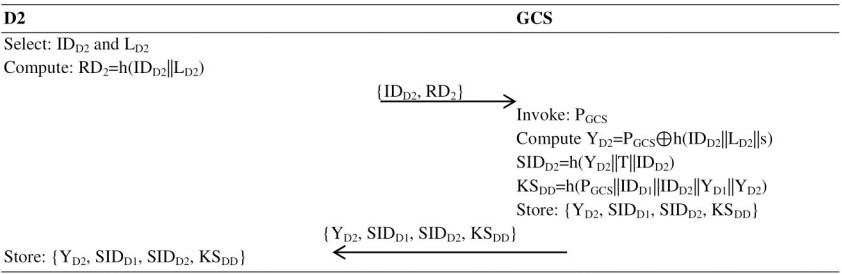

**Fig 3. Registration of the second drone (D2) with the GCS.**

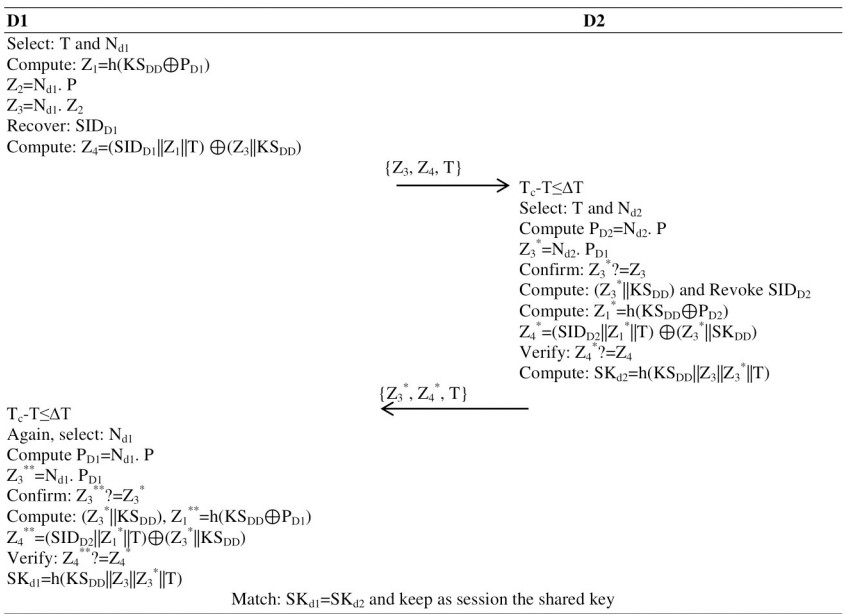

**Fig 4. Drone-to-drone mutual authentication.**

**Step 2:** After receiving the $\{Z_3, Z_4, T\}$ message, the second drone validates the time stamp $T_c$-$T \leq \Delta T$, selects time $T$ and nonce $N_{d2}$, and computes ECC-key $P_{D2} = N_{d2}. P$, $Z_3^* = N_{d2}. P_{D1}$, confirms $Z_3^*? = Z_3$, if validated, computes $(Z_3^*||KS_{DD})$, revokes $SID_{D2}$, computes $Z_1^* = h(KS_{DD} \oplus P_{D2})$, $Z_4^* = (SID_{D2}||Z_1^*||T) \oplus (Z_3^*||SK_{DD})$, verifies $Z_4^*? = Z_4$, if matched, computes the shared session key $SKd2 = h(KS_{DD}||Z_3||Z_3^*||T)$ and sends $\{Z_3^*, Z_4^*\}$ back to the first drone.

**Step 3:** The first drone, after receiving the $\{Z_1^*, Z_4^*, T\}$ message, validates the time stamp $T_c$-$T \leq \Delta T$, selects time and nonce $N_{d1}$ and again calculates the ECC key $Z_3^{**} = N_{d1}. P_{D1}$ where $P_{D1} = N_{d1}. P$, confirms $Z_3^{**}? = Z3^*$ if validated, builds $(Z_3^*||KS_{DD})$, computes $Z_1^{**} = h(KS_{DD} \oplus P_{D1})$, $Z_4^{**} = (SID_{D2}||Z_1^{**}||T)$, verifies $Z_4^{**}? = Z_4^*$, and if matched, computes the session key $SKd1 = h(KS_{DD}||Z_3||Z_3^*||T)$. Now, both drones have mutually authenticated each other with the permission of the GCS, as shown in Fig 4.

## Analysis and discussions

This section provides an analysis and discussion of the proposed drone-to-drone authentication protocol, which can be performed via worldwide techniques. These are explained as follows:

### Security analysis through BAN logic

The security analysis of the proposed scheme can be conducted via the well-known and widely used authentication logic called BAN [43]. The statements are shown in Table 4.

**Message Meaning Rule:** This means that if D1 believes the shared secret among D1 and D2 through key Z3 and sees the encryption of Message M via key K, then D1 believes D2 once

**Table 4. Statements and their pronunciations.**

| Statements | Pronounced as |
|---|---|
| $A \mid\sim B$ | A *Once Said* B |
| $A \lhd B$ | A *Sees* B |
| $<X>$ | Combine |
| $A\mid\equiv B$ | A *Belief* B |
| $A \Rightarrow X$ | A *Jurisdiction* over X |
| $\#(X)$ | X is *Fresh* |
| $(X)$ | Hashing |
| $\{X\}_K$ | Encryption |
| $\{X\}_{K^{-1}}$ | Decryption |
| $A \xleftrightarrow{SK} B$ | An established secure Communication with B through shared session key SK |

Said Message M.

$$\frac{D1\mid \equiv D1\overset{Z3}{\leftrightarrow}ND2\lhd<M>_K}{D1\mid \equiv D2\mid\sim M} \tag{R1}$$

**Verification rule:** This means that if D1 believes the shared secret among D1 and D2 through key Z3 and sees the encryption of M via key K, then D1 believes D2 and has complete jurisdiction over Message M.

$$\frac{D1\mid \equiv D1\overset{Z3}{\leftrightarrow}D2\lhd<M>_K}{D1\mid \equiv D2 \Rightarrow M} \tag{R2}$$

**Freshness rule:** If party D1 believes the freshness of Message M, then D2 also believes the freshness of M

$$\frac{D1\mid \equiv \#(M)}{D2\mid \equiv \#(M)} \tag{R3}$$

**Jurisdiction rule:** If D1 believes the freshness of M and D1 believes D2 has jurisdiction over M, then D1 believes D2 believes the full jurisdiction over Message M.

$$\frac{D1\mid \equiv \#(M), D1\mid \equiv D2\mid\sim M}{D1\mid \equiv D2\mid \equiv\Rightarrow M} \tag{R4}$$

**Message contents.** Drone 1 believes the transmission of message $\{Z_3, Z_4\}$ between drone 1 and drone 2

$$D1\mid \equiv D1 \rightarrow \{Z_3, Z_4\} \rightarrow D2 \tag{MC1}$$

Drone 2 believes the transmission of message $\{Z_3, Z_4\}$ between drone 1 and drone 2

$$D2\mid \equiv D1 \rightarrow \{Z_3, Z_4\} \rightarrow D2 \tag{MC2}$$

Drone 2 believes the transmission of message $\{Z_3{}^*, Z_4{}^*\}$ between drone 2 and drone 1

$$D2\mid \equiv D2 \rightarrow \{Z_3{}^*, Z_4{}^*\} \rightarrow D1 \tag{MC4}$$

Drone 1 believes the transmission of message $\{Z_3{}^*, Z_4{}^*\}$ between drone 2 and drone 1

$$D1| \equiv D2 \rightarrow \{Z_3{}^*, Z_4{}^*\} \rightarrow D1 \tag{MC5}$$

**Idealization.** Drone 1 believes drone 2 believes the nonce they have generated

$$D1| \equiv D2| \equiv N_{d1} \tag{I1}$$

Drone 1 believes drone 2 believes the ECC point they have chosen from the curve

$$D1| \equiv D2| \equiv P_{D1} \tag{I2}$$

Drone 1 believes drone 2 believes the hash code and public key constituted

$$D1| \equiv D2| \equiv Z_3 \tag{I3}$$

Drone 2 believes drone 1 believes the nonce they have generated

$$D2| \equiv D1| \equiv N_{d1} \tag{I4}$$

Drone 2 believes drone 1 believes the hash codes and ECC key

$$D2| \equiv D1| \equiv Z_3 \tag{I5}$$

Drone 2 believes drone 1 the nonce generated

$$D2| \equiv D1| \equiv N_{d2} \tag{I6}$$

Drone 2 believes drone 1 the ECC point chosen

$$D2| \equiv D1| \equiv P_{D2} \tag{I7}$$

Drone 2 believes drone 1 the message and all its credentials

$$D2| \equiv D1| \equiv Z_3{}^* \tag{I8}$$

Drone 1 believes drone 2 believes the nonce of drone 2

$$D1| \equiv D2| \equiv N_{d2} \tag{I9}$$

Drone 1 believes drone 2 the message and all its credentials

$$D1| \equiv D2| \equiv Z_3{}^* \tag{I10}$$

**Goals.**

$$D1| \equiv (D1 \xleftrightarrow{\{Z_3, Z_4\}} D2) \tag{G1}$$

$$D2| \equiv (U \xleftrightarrow{\{Z_3, Z_4\}} CS) \tag{G2}$$

$$D1| \equiv D2| \equiv (D1 \xleftrightarrow{\{Z_3, Z_4\}} D2) \tag{G3}$$

$$D2| \equiv (D2 \xleftrightarrow{\{Z_3^*, Z_4^*\}} D1) \tag{G4}$$

$$D1| \equiv (D2 \xleftrightarrow{\{Z_3^*, Z_4^*\}} D1) \tag{G5}$$

$$D2| \equiv D1| \equiv (D2 \xleftrightarrow{\{Z_3^*, Z_4^*\}} D1) \tag{G6}$$

**1. Assumptions.**

$$D1| \equiv D2 \Rightarrow \{Z_3, Z_4\} \tag{A1}$$

$$D2| \equiv D1 \Rightarrow \{Z_3, Z_4\} \tag{A2}$$

$$D2| \equiv D1 \Rightarrow \{Z_3^*, Z_4^*\} \tag{A3}$$

$$D1| \equiv D2 \Rightarrow \{Z_3^*, Z_4^*\} \tag{A4}$$

$$D1| \equiv SK| \sim \{Z_3, Z_4\} \tag{A5}$$

$$D2| \equiv SK| \sim \{Z_3, Z_4\} \tag{A6}$$

$$D2| \equiv SK| \sim \{Z_3^*, Z_4^*\} \tag{A7}$$

$$D1| \equiv SK| \sim \{Z_3^*, Z_4^*\} \tag{A8}$$

$$D1 \triangleleft Z_3^* \text{ and } D2 \triangleleft Z_3 \tag{A9}$$

$$D2 \triangleleft P_{D2} \text{ and } D1 \triangleleft P_{D2} \tag{A10}$$

$$D1 \triangleleft N_{d2} \text{ and } D2 \triangleleft N_{d1} \tag{A11}$$

$$D1| \equiv \#(N_{d1}) \text{ and } D_2| \equiv \#(N_{d1}) \tag{A12}$$

$$D1| \equiv \#(Z_3, Z_4) \tag{A13}$$

$$D2| \equiv \#((Z_3, Z_4) \tag{A14}$$

$$D1| \equiv \#(Z_3^*, Z_4^*) \tag{A15}$$

$$D2| \equiv \#(Z_3^*, Z_4^*) \tag{A16}$$

$$D1| \equiv (D1 \overset{SK}{\leftrightarrow} D2) \tag{A17}$$

$$D2| \equiv (D1 \overset{SK}{\leftrightarrow} D2) \tag{A18}$$

$$D1| \equiv D2| \equiv (D1 \overset{SK}{\leftrightarrow} D2) \tag{A19}$$

$$D2| \equiv (D2 \overset{SK}{\leftrightarrow} D1) \tag{A20}$$

$$D1| \equiv (D2 \overset{SK}{\leftrightarrow} D1) \tag{A21}$$

$$D2| \equiv D1| \equiv (D2 \overset{SK}{\leftrightarrow} D1) \tag{A22}$$

**Proof.** The message contents, idealizations and assumptions are used to prove the goals. Taking MC1, I1, and A1, we obtain

$$D1| \equiv D2| \equiv N_{d1} \Rightarrow \{Z_3, Z_4\}| \equiv D1 \rightarrow \{Z_3, Z_4\} \rightarrow D2 \tag{1}$$

Eqs (1), (I7) and (A7) yield

$$D1| \equiv D2| \equiv N_{d1}, P_{D2}|\sim \{Z_3, Z_4\} \Rightarrow D1 \rightarrow \{Z_3, Z_4\} \rightarrow D2 \tag{2}$$

Eqs (2), (I9), (I10) and (A15) yield

$$D1| \equiv D2| \equiv \#(N_{d1}, P_{D2})| \sim \{Z_3, Z_4\} \Rightarrow D1 \rightarrow \{Z_3, Z_4\} \rightarrow D2 \tag{3}$$

Eqs (3), (MC2) and (A17)

$$D1| \equiv D2| \equiv \#(N_{d1}, P_{D2})| \sim \{Z_3, Z_4\}| \equiv (D1 \overset{SK}{\leftrightarrow} D2) \tag{4}$$

Eq (4) can be written as

$$D1| \equiv D2| \equiv \#(N_{d1}, P_{D2})| \equiv D1 \xleftarrow{\{Z_3, Z_4\}} D2) \tag{5}$$

Eqs (5), (I2), (I3) and (A12)

$$D1| \equiv \#(N_{d1})| \equiv D1 \xleftarrow{\{Z_3, Z_4\}} D2) \tag{6}$$

and

$$D2| \equiv \#(P_{D2})| \equiv D1 \xleftrightarrow{\{Z_3, Z_4\}} D2) \tag{7}$$

Eqs (6) and (7) can be expressed as

$$D1| \equiv (D1 \xleftrightarrow{\{Z_3, Z_4\}} D2) \tag{G1 — Achieved}$$

$$D2| \equiv (D1 \xleftrightarrow{\{Z_3, Z_4\}} D2) \tag{G2 — Achieved}$$

According to G1, G2, and Eq (5), the result obtained as follows:

$$D1| \equiv D2| \equiv (D1 \xleftrightarrow{\{Z_3, Z_4\}} D2) \tag{G3 — Achieved}$$

Taking MC2, I4, and A3, we obtain

$$D2| \equiv D1| \equiv N_{d1} \Rightarrow \{Z_3, Z_4\}| \equiv D2 \rightarrow \{\{Z_3^*, Z_4^*\} \rightarrow D1 \tag{8}$$

Eqs (8), (I5) and (A8) yield

$$D2| \equiv D1| \equiv N_{d1}, Z_3| \sim \{Z_3^*, Z_4^*\} \Rightarrow D2 \rightarrow \{Z_3^*, Z_4^*\} \rightarrow D1 \tag{9}$$

Eqs (9), (I6), (I7) and (A16) yield

$$D2| \equiv D1| \equiv \#(N_{d1}, Z_3)| \sim \{Z_3^*, Z_4^*\} \Rightarrow D2 \rightarrow \{Z_3^*, Z_4^*\} \rightarrow D1 \tag{10}$$

Eqs (10), (MC4) and (A18)

$$D2| \equiv D1| \equiv \#(N_{d1}, Z_3, N_{d2})| \sim \{Z_3^*, Z_4^*\}| \equiv (D2 \xleftrightarrow{SK} D1) \tag{11}$$

Eq (11) can be written as

$$D2| \equiv D1| \equiv \#(N_{d1}, Z_3, N_{d2})| \equiv D2 \xleftrightarrow{\{Z_3^*, Z_4^*\}} D1) \tag{12}$$

Eqs (12), (I8), (I9) and (A12)

$$D2| \equiv \#(N_{d1})| \equiv D2 \xleftrightarrow{\{Z_3^*, Z_4^*\}} D1) \tag{13}$$

and

$$D2| \equiv \#(Z_3)| \equiv D2 \xleftrightarrow{\{Z_3^*, Z_4^*\}} D1) \tag{14}$$

and

$$D2| \equiv \#(P_{D1})| \equiv D2 \xleftrightarrow{\{Z_3^*, Z_4^*\}} D1) \tag{15}$$

Eqs [(13)](13), [(14)](14) and [(15)](15) can be expressed as

$$D2| \equiv (D2 \xleftrightarrow{\{Z_3^*, Z_4^*\}} D1) \qquad (G4 \text{— Achieved})$$

$$D1| \equiv (D2 \xleftrightarrow{\{Z_3^*, Z_4^*\}} D1) \qquad (G5 \text{— Achieved})$$

According to G4, G5, and [Eq (12)](12), the result obtained as follows:

$$D2 \equiv D1 \equiv (D2 \xleftrightarrow{\{Z_3^*, Z_4^*\}} D1) \qquad (G6 \text{— Achieved})$$

## Security analysis through ProVerif

ProVerif [44], a worldwide verification software toolkit, verifies the session key's secrecy, confidentiality, authorization, and reachability. In this context, we first define two channels, i.e., private and public, and then define all the variables, constraints, constants, equations, and queries. We then code each computation step and run the code to check for weak points as presented in separate file. If none are found, we obtain the verification summary result, which shows that the attacker cannot crack the session key at any stage and that the MITM attack is not possible on the proposed scheme.

```
--------------------------------------------------
Verification summary:
The query, not attacker(skd1[]), is true.
The query, not attacker(skd2[]), is true.
The query inj-event(DroneEnd(id)) ==> inj-event(DroneStart
(id)) is true.
--------------------------------------------------
```

## Informal analysis

This section presents an analysis of the proposed D2D mutual authentication protocol through a pragmatic discussion of different attacks.

**Anonymity.** The message transmitted from D1 to D2 is $\{Z_3, Z_4, T\}$, which includes $Z_3 = N_{d1}$. $Z_2$ whereas $Z_2 = N_{d1}$. P and $Z_4 = (SID_{D1}||Z_1||T) \oplus (Z_3||KS_{DD})$, and the message from D2 to D1 is $\{Z_3^*, Z_4^*, T\}$, which includes $Z_3^* = N_{d2}$. $P_{D1}$ and $Z_4^* = (SID_{D2}||Z_1^*||T) \oplus (Z_3^*||SK_{DD})$. All the messages are concealed from an attacker so that the location of a drone cannot be identified. If an attacker receives a message from the public channel, they do not succeed in obtaining any credentials, and a legitimate drone maintains its anonymity.

**Replay attack.** The first drone selects a timestamp and nonce $N_{d1}$ and computes $Z_1 = h(KS_{DD} \oplus P_{D1})$, $Z_2 = N_{d1}$. P, $Z_3 = N_{d1}$. $Z_2$ retrieves $SID_{D1}$ and computes $Z_4 = (SID_{D1}||Z_1||T) \oplus (Z_3||KS_{DD})$, builds $\{Z_3, Z_4, T\}$ and sends it to D2. If an attacker obtains $\{Z_3, Z_4, T\}$ from the open network channel, they have to pass from $T_c\text{-}T \leq \Delta T$ and through many other checks, such as $Z_3^*? = Z_3$ and $Z_4^*? = Z_4$, which is impossible. Similarly, the second drone selects the timestamp T and nonce $N_{d2}$ and computes $P_{D2} = N_{d2}$. P, $Z_3^* = N_{d2}$. $P_{D1}$, $Z_1^* = h(KS_{DD} \oplus P_{D2})$, $Z_4^* = (SID_{D2}||Z_1^*||T) \oplus (Z_3^*||SK_{DD})$, $SKd2 = h(KS_{DD}||Z_3||Z_3^*||T)$ builds $\{Z_3^*, Z_4^*, T\}$ and sends it to the first drone. Suppose that an attacker obtains the $\{Z_3^*, Z_4^*, T\}$ message from the wireless communication channel and tries to inflict a replay attack. In this case, the attacker must validate $T_c\text{-}T \leq \Delta T$ and perform many other random checks, such as $Z_3^{**}? = Z_3^*$ and $Z_4^{**}? = Z_4^*$, which is impossible. Therefore, the proposed protocol is resistant to replay attacks.

**Insider attack.** The credentials in the memory of the GCS include $\{Y_{D1}, SID_{D1}, SID_{D2}, KS_{DD}\}$ credentials consisting of $Y_{D1} = P_{GCS} \oplus h(ID_{D1}||L_{D1}||s)$, $SID_{D1} = h(Y_{D1}||T||ID_{D1})$, and $KS_{DD} = h(P_{GCS}||ID_{D1}||ID_{D2}||Y_{D1}||Y_{D2})$. Furthermore, $\{Y_{D2}, SID_{D1}, SID_{D2}, KS_{DD}\}$ credentials consist of $Y_{D2} = P_{GCS} \oplus h(ID_{D2}||L_{D2}||s)$, $SID_{D2} = h(Y_{D2}||T||ID_{D2})$, and $KS_{DD} = h(P_{GCS}||ID_{D1}|| ID_{D2}||Y_{D1}||Y_{D2})$. The GCS also stores $\{E(x, y), P, P_{GCS}, h(.)\}$ and secret keys, which are 160- and 64-bit keys, respectively. Suppose that an attacker enters the GCS and acts as an insider attacker, as the GCS does not have a database table. In this case, the attacker will not be able to find anything, so their attempt will fail due to the availability of useful information in encrypted form. Therefore, the proposed D2D mutual authentication scheme is resistant to insider attacks.

**Key disclosure attack.** The memory of the second drone consists of a shared secret key $KS_{DD}$ of the GCS. It selects the timestamp T and nonce $N_{d2}$ and computes $P_{D2} = N_{d2}. P$, $Z_3^* = N_{d2}. P_{D1}$, confirms $Z_3^*? = Z_3$ and computes $(Z_3^*||KS_{DD})$ and revokes the pseudo identity $SID_{D2}$ and computes $Z_1^* = h(KS_{DD} \oplus P_{D2})$, $Z_4^* = (SID_{D2}||Z_1^*||T) \oplus (Z_3^*||SK_{DD})$, verifies $Z_4^*? = Z_4$ and computes the session secret key $SK_{d2} = h(KS_{DD}||Z_3||Z_3^*||T)$. Similarly, the first drone selects $N_{d1}$ and computes $P_{D1} = N_{d1}. P$, $Z_3^{**} = N_{d1}. P_{D1}$ confirms $Z_3^{**}? = Z_3^*$ and computes $(Z_3^*||KS_{DD})$, $Z_1^{**} = h(KS_{DD} \oplus P_{D1})$, $Z_4^{**} = (SID_{D2}||Z_1^*||T) \oplus (Z_3^*||KS_{DD})$, verifies $Z_4^{**}? = Z_4^*$ and computes the session secret key $SK_{d1} = h(KS_{DD}||Z_3||Z_3^*||T)$, which the attacker cannot disclose because it consists of a complex set of calculations.

**Stolen-verifier attack.** If an attacker manages to take down or capture a drone and tries to access its internal credentials, they will be unable to do so because they are in encrypted form. The first drone selects an identity $ID_{D1}$ and random number $L_{D1}$, calculates $RD_1 = h(ID_{D1}||L_{D1})$ and sends $\{ID_{D1}, RD_1\}$ to GCS, where it invokes its public key $P_{GCS}$, computes $Y_{D1} = P_{GCS} \oplus h(ID_{D1}||L_{D1}||s)$ and calculates the pseudoidentity for the first drone, $SID_{D1} = h(Y_{D1}. T||ID_{D1})$ and pseudoidentity for the second drone, $SIDD2 = h(YD2. T||IDD2)$ and builds a long-term shared secret key $KSDD = h(_{PGCS||YDD1||IDD2||YD1||YD2})$ and sends $\{Y_{D1}, SID_{D1}, SID_{D2}, KS_{DD}\}$ to the first drone and second drone, which they store in their memory $\{Y_{D1}, SID_{D1}, SID_{D2}, KS_{DD}\}$. Hence, these stored credentials are secure, and an attacker will be unable to learn anything. Therefore, the proposed D2D mutual authentication protocol is resistant to stolen-verifier attacks.

**Forward secrecy.** If any change is made to the parameters, it securely changes all the corresponding parameters because they are cross-connected. The proposed D2D mutual authentication protocol, which incorporates forward secrecy, ensures that attackers cannot compromise sensitive information.

## Performance analysis

When measured in terms of communication and computation costs, the proposed protocol's performance metrics yield significant results. In [44–47], the memory space occupied by the ECC key, nonce, timestamp, identity, and SHA-1 is 160 bits, 60 bits, 32 bits, 64 bits, and 256 bits, respectively. According to [44, 47], the execution time for the one-way hash ($T_{hash}$) cryptographic function is 0.0046 ms, the ECC point multiplication ($T_{Mul}$) is 2.226 ms, the ECC point addition ($T_{Add}$) is 0.0288 ms, and the XOR and concatenation functions are zero, which should be avoided. This caution about using certain functions is important for the audience to note. Based on these results, the communication costs are the publicly exchanged messages between D1 and D2, i.e., $\{Z_3, Z_4\}$ and $\{Z_3^*, Z_4^*\}$, resulting in a communication cost of 1344 bits. Similarly, the computational cost of the proposed protocol is $5T_{hash} + 6T_{Mul} + 3T_{Add}$, resulting in a cost of 13.5 ms.

**Table 5. Comparative analysis in terms of performance metrics.**

| Schemes→ Performance Metrics↓ | [31] | [38] | [37] | [41] | [42] | [33] | [48] | [49] | Proposed |
|---|---|---|---|---|---|---|---|---|---|
| Communication Costs | 3720 | 2728 | 4320 | 3232 | 3264 | 1664 | 3040 | 4432 | 1344 |
| Computation Costs | 17.79 | 31.158 | 14.08 | 12.447 | 40.211 | 9.54 | 15.14 | 36.171 | 13.5 |

We are now comparing the proposed protocol with Jan et al. [31], Amin et al. [38], Alzahrani [41], Jan et al. [42], Algarni et al. [33], Irshad et al. [48], and Tanveer et al. [49] and underscoring the significant superiority of the proposed protocol. It not only has lower communication costs than all of the mentioned schemes but also surpasses all the mentioned schemes in terms of efficiency. The computational costs of [33, 41] may be slightly lower than those of the proposed protocol, but their communication costs are unacceptably high, as shown in Table 5 and plotted in Figs 5 and 6.

It is important to note that the computational costs of the proposed protocol, while higher, are within acceptable limits, ensuring the feasibility of the protocol. This assurance of practicality is crucial for the reader. For example, Jan et al. [31]'s method for securing the IoD environment has communication costs of 3720 bits and computational costs of 17.79 milliseconds. In contrast, their work in [42] has communication costs of 2728 bits and computational costs of 31.158 milliseconds. Similarly, the authentication scheme presented by Amin et al. [38] has

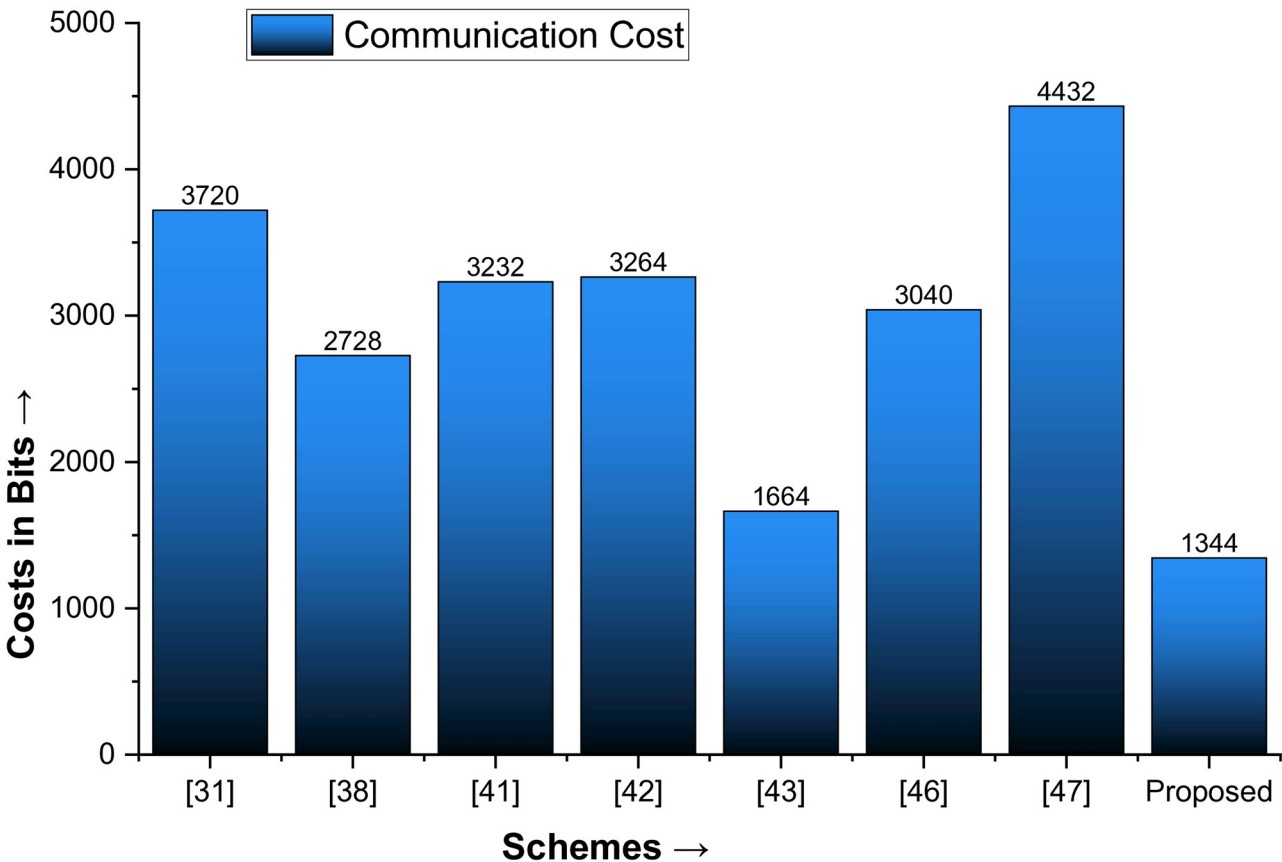

**Fig 5. Comparison of the communication cost with state-of-the-art schemes.**

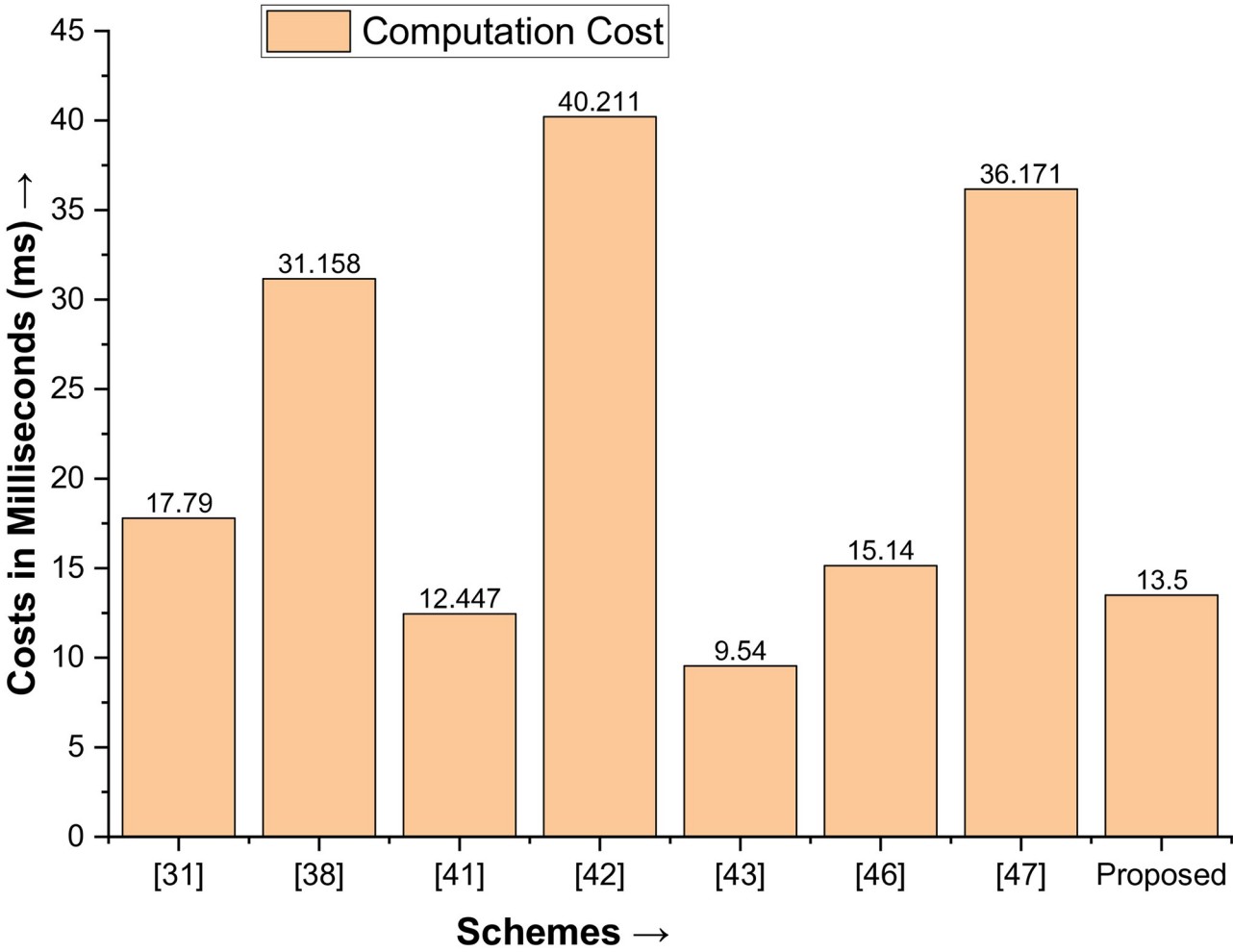

**Fig 6. Comparison of the computation cost with state-of-the-art schemes.**

a communication cost of 4320 bits and a computational cost of 14.08 milliseconds, and the method proposed by Alzahrani [41] has a communication cost of 3232 bits and a computational cost of 12.447 milliseconds. Even compared with the communication costs of Algarni et al. [33], Irshad et al. [48], and Tanveer et al. [44], which are 3264-bit, 1664-bit, 3040-bit, and 4432-bit are higher than the proposed protocol, the efficiency of the proposed protocol remains unmatched. In conclusion, the D2D mutual authentication protocol's significant superiority over its competitors is undeniable.

The proposed protocol significantly outperforms existing ones in terms of performance, with impressive improvement percentages. It surpasses [31] by 63.87% in communication costs and 24.41% in computational costs. Compared to [42], the proposed scheme is 50.73% faster and 56.84% more efficient. In the case of [38], the proposed protocol is 48.88% better in communication costs and 4.49% in computation costs. It also outshines [41] by 58.41% in communication costs. While the computational costs of [41] are slightly lower than those of the proposed protocol, the overall superiority of the proposed protocol is evident. The communication cost presented by Tanveer et al. [49] is 4432 bit, which is also higher than the proposed scheme, making the proposed scheme 69.67% better than the Tanveer et al. [49] scheme in terms of communication costs as shown in Table 6 and plotted in Fig 7.

**Table 6. Percentage improvement of the proposed scheme over state-of-the-art schemes.**

| Scheme | % Improvement in Communication Cost | % Improvement in Computation Cost |
|---|---|---|
| Jan et al. [31] | 63.87% | 24.41% |
| Amin et al. [38] | 48.88% | 4.49% |
| Gutub et al. [49] | 50.73% | 56.84% |
| Alzahrani [41] | 58.41% | NIL |
| Jan et al. [42] | 50.73% | 56.84% |
| Algarni et al. [33] | 58.82% | 66.42% |
| Irshad et al. [48] | 55.78% | 10.83% |
| Tanveer et al. [49] | 69.67% | 62.67% |

Similarly, the communication and computation costs of the method proposed in [43] are 3264 bits and 40.211 milliseconds, respectively. In contrast, the proposed protocol offers reduced costs of 1344 bits, a significant 58.82% less than that of [41], and 13.5 milliseconds, which is a substantial 66.42% better than that of [41]. While the computational cost of [42] is marginally better than that of the proposed protocol, its communication cost is higher, and

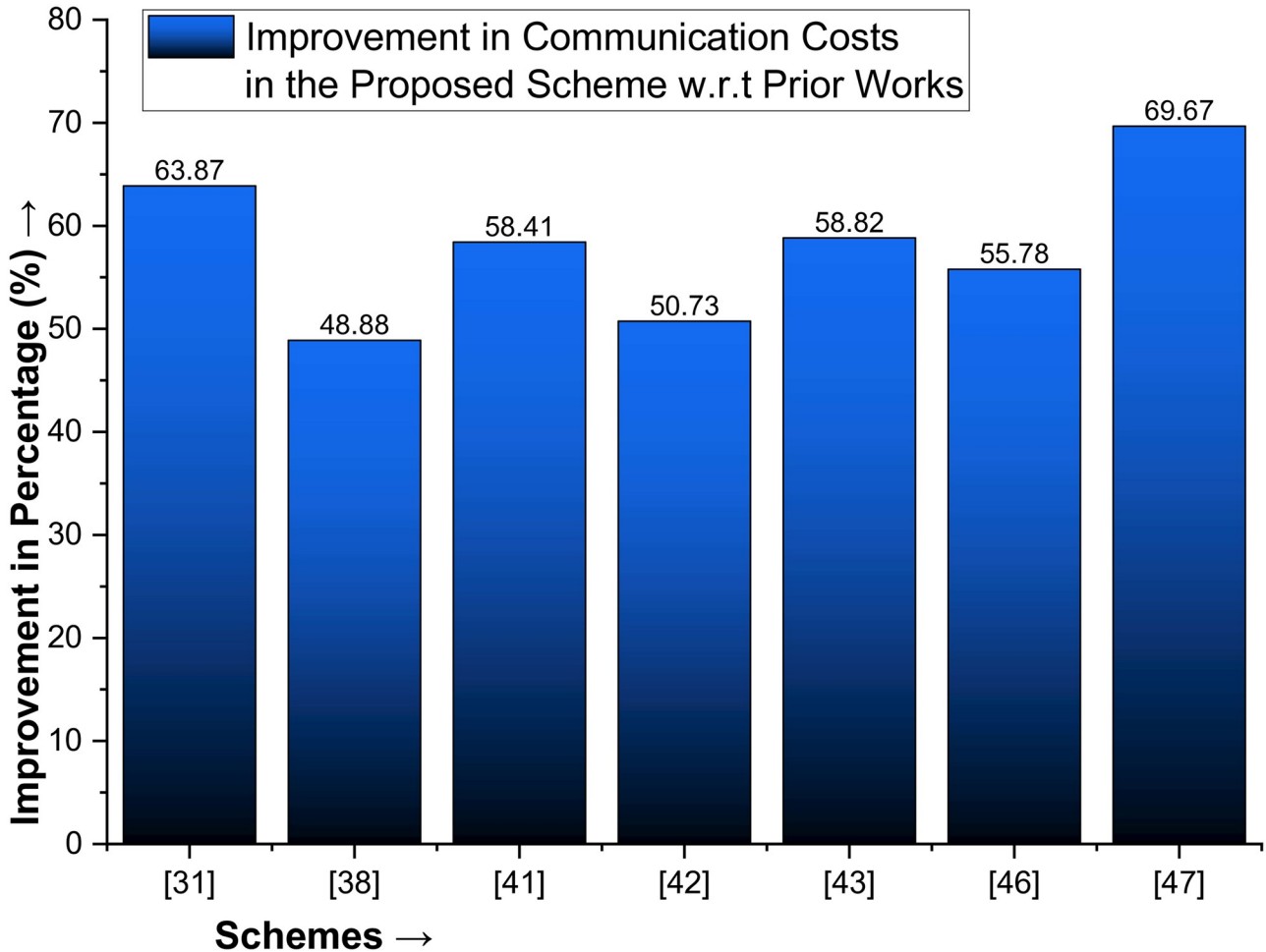

**Fig 7. Improvement in communication cost for the proposed scheme with state of the art protocols.**

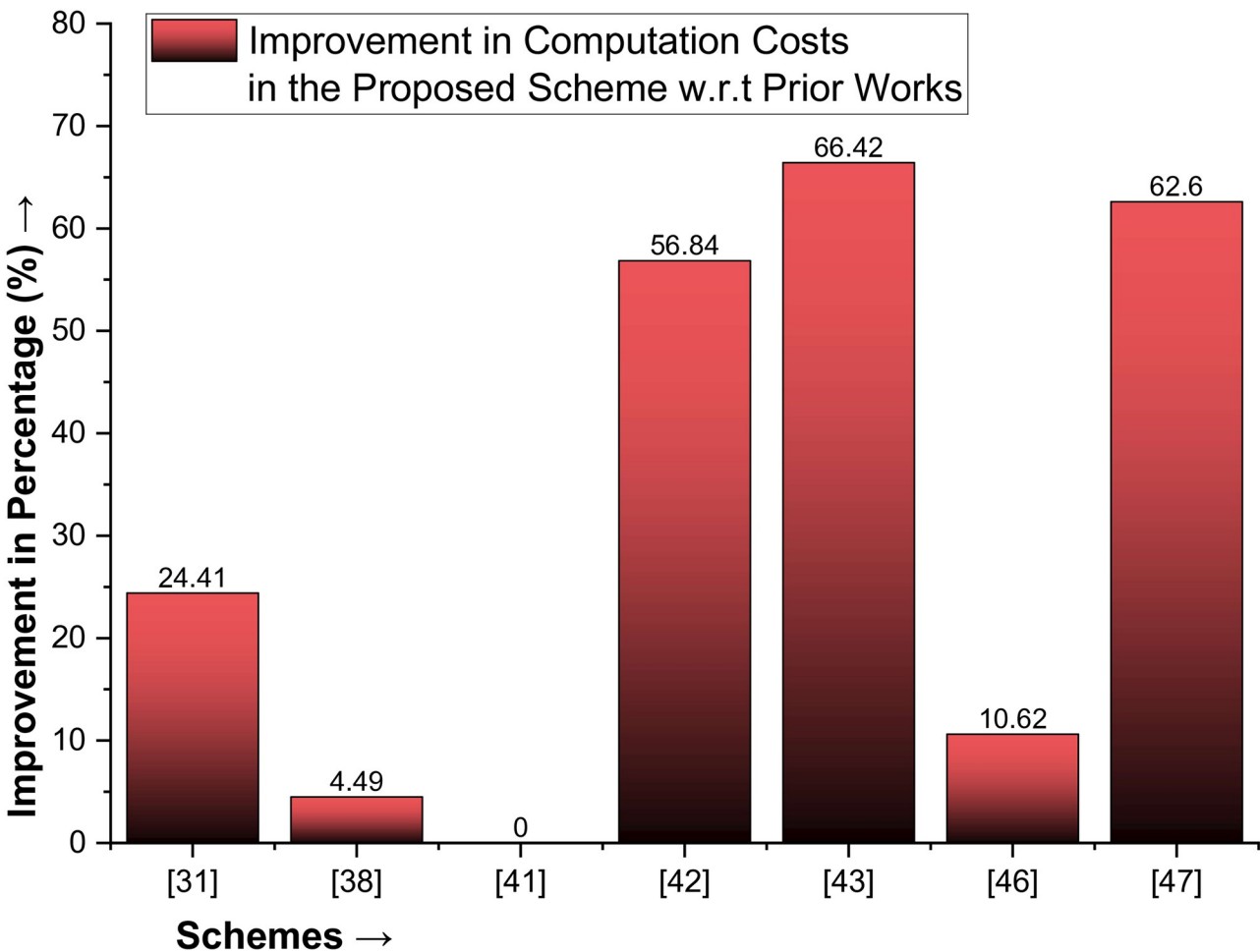

**Fig 8. Improvement in communication cost for the proposed scheme with state of the art protocols.**

our scheme is 16.23% better. A comparison of the proposed protocol with [33] demonstrated its superiority in communication and computation costs, with percentages of improvement of 55.78% and 10.83%, respectively. Additionally, the computation cost of Tanveer et al. [49] is 36.171 ms, and the proposed scheme is 62.67% better in computation cost than [49]. Most importantly, the proposed protocol effectively balances security with performance, a crucial aspect often missing in prior works, thereby instilling confidence in its effectiveness and providing a sense of reassurance, as shown in Table 6 and plotted in Fig 8.

In this paragraph, we present a comprehensive comparison with prior works regarding the security characteristics, functionalities, and features of the proposed D2D mutual authentication protocol. The proposed D-to-D protocol provides more robust security than the four current techniques, specifically Jan et al. [31], Amin et al. [38], Alzahrani [41], Jan et al. [42], Algarni et al. [33], Irshad et al. [48], and Tanveer et al. [49], as demonstrated in Table 7.

One problem with the work proposed in [31] is that a privileged insider attacker can create session keys with every other drone if they can access the identities of every registered drone. A problem with the work proposed in [42] is that the attacker can independently compute the session key if they register with the GCS for a previous session, which means there will be no security for the next session. However, the involvement of the GCS is no longer required in

**Table 7. Comparative analysis in terms of security functionalities.**

| Schemes→ Security Functionalities↓ | [31] | [38] | [37] | [41] | [42] | [33] | [48] | [49] | Proposed |
|---|---|---|---|---|---|---|---|---|---|
| Man-in-the-Middle Attack | ✗ | ✓ | ✓ | ✗ | ✗ | ✓ | ✗ | ✗ | ✗ |
| Replay Attack | ✗ | ✗ | ✗ | ✓ | ✓ | ✗ | ✓ | ✗ | ✗ |
| DoS Attack | ✗ | ✗ | ✓ | ✗ | ✗ | ✓ | ✗ | ✗ | ✗ |
| Side Channel Attack | ✗ | ✗ | ✗ | ✓ | ✗ | ✗ | ✓ | ✗ | ✗ |
| Insider Attack | ✓ | ✗ | ✗ | ✓ | ✗ | ✗ | ✓ | ✓ | ✗ |
| Stolen-Verifier Attack | ✗ | ✗ | ✓ | ✗ | ✓ | ✓ | ✗ | ✗ | ✗ |
| ESL Attack | ✗ | ✓ | ✗ | ✗ | ✓ | ✗ | ✗ | ✗ | ✗ |
| Anonymity | ✓ | ✓ | ✗ | ✗ | ✓ | ✗ | ✗ | ✓ | ✓ |
| Untraceability | ✓ | ✓ | ✗ | ✗ | ✓ | ✗ | ✗ | ✓ | ✓ |
| Forward Secrecy | ✓ | ✓ | ✗ | ✓ | ✓ | ✗ | ✓ | ✓ | ✓ |
| Session Key Disclosure Attack | ✓ | ✓ | ✗ | ✓ | ✓ | ✗ | ✓ | ✗ | ✗ |
| Impersonation Attack | ✗ | ✗ | ✓ | ✓ | ✗ | ✓ | ✓ | ✗ | ✗ |
| Mutual Authentication | ✓ | ✗ | ✗ | ✗ | ✗ | ✗ | ✗ | ✓ | ✓ |
| Backward Secrecy | ✓ | ✓ | ✗ | ✗ | ✓ | ✗ | ✗ | ✓ | ✓ |

✓ means that the specified vulnerability/feature exists

✗ means the specified attack/feature doesn't exist.

our scenario, highlighting the efficiency of our proposed D2D protocol and offering a promising solution. The inability to obtain session key agreement is an issue that arises in the work proposed in [38]. Furthermore, crucial data, such as long-term secret keys for the subsequent session, are kept in the drone's memory in work proposed by Jan et al. [31, 37], Amin et al. [38], and Alzahrani [41], making the protocols susceptible to physical threats. When the proposed method is compared with the methods in [48, 49], the results show that the proposed D2D protocol has higher security requirements than the previous protocols do. Additionally, [49] is weak against insider attacks while safe against all attacks, and all the features are shown in Table 7; overall, our scheme is superior to its competitors.

## Conclusion

This article presents a synergistic-assisted authentication protocol for the IoD environment, a context where drones are utilized for complex task completion. The protocol, designed using elliptic curve cryptography (ECC), a one-way hash cryptography function, and XOR operations, ensures the security of drone-to-done communication. Each drone is first registered with the ground control station (GCS) offline via a secure channel and then deployed in the IoD environment. However, a real-time secure exchange of information is crucial, necessitating rigorous authentication, which is the focus of this article. The security of the proposed protocol is thoroughly analyzed via two methods, i.e., BAN logic and ProVerif simulation, instilling confidence in its effectiveness. Its performance is evaluated in terms of communication and computation costs. Upon comparison with recent schemes from the perspective of both performance and security, it has been demonstrated that the proposed approach is robust and lightweight, ensuring secure communication in the real world for drone technology. Looking ahead, the protocol will be tested via the Scyther tool, random oracle (ROM), and real-or-random (ROR) models, and operationalized for the Quantum Key Distribution (QKD) method using a quantum key distribution network, offering a promising future for secure drone communication.

## Supporting information

**S1 File. ProVerif simulation code.**
(TXT)

## Acknowledgments

Ali Algarni and Fahad Algarni are thankful to the University of Bisha, Saudi Arabia, for their support and encouragement, while the second author, Nisreen Innab, would like to express sincere gratitude to AlMaarefa University, Riyadh, Saudi Arabia.

## Author Contributions

**Conceptualization:** Ali Delham Algarni.

**Data curation:** Ali Delham Algarni.

**Formal analysis:** Nisreen Innab.

**Methodology:** Nisreen Innab.

**Project administration:** Ali Delham Algarni.

**Resources:** Ali Delham Algarni.

**Supervision:** Ali Delham Algarni.

**Validation:** Nisreen Innab, Fahad Algarni.

**Visualization:** Nisreen Innab, Fahad Algarni.

**Writing – original draft:** Fahad Algarni.

**Writing – review & editing:** Fahad Algarni.

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
