## [Decision Letter · Decision Letter 0]

16 Oct 2024

PONE-D-24-39087A Verifiably Secure and Robust Authentication Protocol for Synergic-Assisted IoD Deployment DronesPLOS ONE

Dear Dr. Algarni,

Thank you for submitting your manuscript to PLOS ONE. After careful consideration, we feel that it has merit but does not fully meet PLOS ONE’s publication criteria as it currently stands. Therefore, we invite you to submit a revised version of the manuscript that addresses the points raised during the review process.

While the paper presents a potentially valuable contribution, I believe it requires significant revisions to reach a publishable standard. 

Improve the structure and clarity of the introduction, clearly outlining the problem statement and objectives.

Update the literature review with recent references and critically analyze existing protocols.

A more detailed response to each of the reviewers' comments and a revised manuscript that incorporates these changes will be necessary for further consideration. 

We look forward to receiving your revised manuscript.

Kind regards,

Md Tarique Jamal Ansari, Ph.D.

Academic Editor

PLOS ONE

“The authors thank the Deanship of Scientific Research at the University of Bisha, Saudi Arabia, for funding this research, and Dr. Nisreen Innab would like to express sincere gratitude to AlMaarefa University, Riyadh, Saudi Arabia, for supporting this research.”

5. We note that Figure 1 in your submission contain copyrighted images. All PLOS content is published under the Creative Commons Attribution License (CC BY 4.0), which means that the manuscript, images, and Supporting Information files will be freely available online, and any third party is permitted to access, download, copy, distribute, and use these materials in any way, even commercially, with proper attribution. For more information, see our copyright guidelines: http://journals.plos.org/plosone/s/licenses-and-copyright.

6. We notice that your supplementary figures are uploaded with the file type 'Figure'. Please amend the file type to 'Supporting Information'. Please ensure that each Supporting Information file has a legend listed in the manuscript after the references list.

Additional Editor Comments:

While the paper presents a potentially valuable contribution, I believe it requires significant revisions to reach a publishable standard.

Improve the structure and clarity of the introduction, clearly outlining the problem statement and objectives.

Update the literature review with recent references and critically analyze existing protocols.

A more detailed response to each of the reviewers' comments and a revised manuscript that incorporates these changes will be necessary for further consideration.

Reviewers' comments:

Reviewer's Responses to Questions

**Comments to the Author**

1. Is the manuscript technically sound, and do the data support the conclusions?

Reviewer #1: Yes

Reviewer #2: Yes

2. Has the statistical analysis been performed appropriately and rigorously? 

Reviewer #1: I Don't Know

Reviewer #2: Yes

3. Have the authors made all data underlying the findings in their manuscript fully available?

Reviewer #1: Yes

Reviewer #2: Yes

4. Is the manuscript presented in an intelligible fashion and written in standard English?

Reviewer #1: Yes

Reviewer #2: Yes

5. Review Comments to the Author

Reviewer #1: This paper presents an interesting utilization of robust and lightweight authentication scheme to handle information fusion and collaboration coordination of drones working in clusters. It aims to be efficient and effective running essential components of limited computing and storage resources serving internet of Drones (IoD) technology. The study adopts elliptic curve cryptography (ECC) and hash as well as exclusive-OR (XOR) operations and concatenation to establish secure communication session between drones run efficiently for complex task completion and information fusion. The security of the proposed protocol has been analyzed via BAN logic and ProVerif trying to stand some common attack in promising progression. The works security and robustness computation and cost are having significance, but the overall research presentation needs to be slightly improved related to others, in order to be ready as publication, as noted within the following points that have to be fulfilled:

1- The abstract needs to be revised a bit and stressing more on specific key contribution and originality. Try reducing general knowledge aiming to attract the reader to select the paper to study and refer to as well as get motivated toward the work to continue research in similar direction. Try reducing answering ‘how’ question within the abstract, i.e. making it focus on ‘what’ question as its main phrasing presentation ?

2- Give more elaboration on the real need for utilizing this elliptic curve crypto authentication proposal among other approaches. What is wrong in the normal other related hash authentication methods requiring this kind of complex research. Try to support your explanation via real-life examples.

3- The IoD security presented is very promising showing its limited resources trade-off to security and robustness improvement as challenging processes. Try to benefit from the light crypto and authentication secrecy researches below linking to the secrecy vs cost complexity attempts provided:

== "Enhancing Speed of SIMON: A Light-Weight-Cryptographic Algorithm for IoT Applications", Multimedia Tools and Applications (MTAP) 78:32633–32657 (2019)

== "Simulating Light-Weight-Cryptography Implementation for IoT Healthcare Data Security Applications", International Journal of E-Health and Medical Communications (IJEHMC) 10(4):1-15 (2019)

== "Integrating Light-Weight Cryptography with Diacritics Arabic Text Steganography Improved for Practical Security Applications", Journal of Information Security and Cybercrimes Research (JISCR) 3(1):13-30 (2020)

== "Secure Mobile Computing Authentication Utilizing Hash, Cryptography and Steganography Combination", Journal of Information Security and Cybercrimes Research (JISCR) 2(1):73-82 (2019)

== “Applicable Light-Weight Cryptography to Secure Medical Data in IoT Systems”, Journal of Research in Engineering and Applied Sciences (JREAS) 2(2):50-58 (2017)

== "Engineering Graphical Captcha and AES Crypto Hash Functions for Secure Online Authentication", Journal of Engineering Research (JER) 11(3): 100-111 (2023)

== "Watermarking Images via Counting-Based Secret Sharing for Lightweight Semi-Complete Authentication",  International Journal of Information Security and Privacy (IJISP) 16(1): 1-18 (2022)

== “Adjusting counting-based secret-sharing via personalized passwords and email-authentic reliability”, Journal of Engineering Research (JER), 12(1): 107-121 (2024)

4- The testing needs more elaboration in different ways. You need to clarify more simple example given showing other possible scenarios with real details stressing on your proposal among others. Note that the example cases need to further consider not being system-confusion in several models as well as being in misleading technical appearance.

5- Combining cryptography and hash is used here as very promising philosophy to stand the IoD trust authenticity privacy system against many security attacks. Although this combination is noted as extra payload in this direction, briefly link your work idea to the complexity of presented IoT similar combinations within the following studies:

== "Efficient computation of Hash Hirschberg protein alignment utilizing hyper threading multi-core sharing technology", CAAI Transactions on Intelligence Technology, IET (IEE) - Wiley, 7(2): 278–291 (2022)

== "Integrity verification for digital Holy Quran verses using cryptographic hash function and compression", Journal of King Saud University - Computer and Information Sciences 32(1):24-34 (2020)

6- The work lack reasoning within given comparisons. It needs many more logical thinking justifications supporting its contrasting to others proofing its real applicability. It also needs elaboration proofing fairness in the results and comparison, especially testing the schemes on variation of similar security IoD schemes. It needs some indication with more circumstances of these results. Explain more the observations feedback of the proposed work vs. others ?

7- This work is elliptic curve crypto procedure for its less computation high resilience security. Remark on the proposal given in light of considering the following ECC encryption confidentiality attempts:

== “Fast 160-Bits GF(p) Elliptic Curve Crypto Hardware of High-Radix Scalable Multipliers”, International Arab Journal of Information Technology (IAJIT) 3(4):342-349

== “Area Flexible GF(2k) Elliptic Curve Cryptography Coprocessor”, International Arab Journal of Information Technology (IAJIT) 4(1):1-10

== "Implementation of a pipelined modular multiplier architecture for GF(p) elliptic curve cryptography computation", Kuwait Journal of Science and Engineering (KJSE) 38(2B):125-153

== "Enhancing Medical Data Security via Combining Elliptic Curve Cryptography with 1-LSB and 2-LSB Image Steganography", International Journal of Computer Science and Network Security (IJCSNS) 20(12):232-241 (2020)

8- The list shown in Table 6 of percentage improvement linking proposed scheme over state-of-the-art schemes as well as comparative analysis of security functionalities of Table 7 for the proposed scheme is kind of misleading as reported needing to justified on similar testing platform and configuration circumstances. It is recommended that these tables be elaborated and justified fair to be acceptable logical attitude.

9- Conclusion needs reconsideration. It needs to highlight the research main contribution with some brief indications and numerical improvement percentages to keep with the reader. Also, the conclusion needs to present some more ideas of open research and future work for researchers to build upon for further advancements.

Reviewer #2: This article presents a protocol based on elliptic curve cryptography (ECC), one-way hash, concatenation, and exclusive-OR (XOR) operations to establish a secure communication session between drones efficiently for complex task completion and efficient information fusion. The security of the proposed protocol has formally been scrutinized via BAN logic and ProVerif, as well as informally through attack discussion. The results obtained from the analysis demonstrate that the proposed protocol is superior and feasible for secure communication. Overall, the proposed protocol is 63.87% lightweight in terms of communication costs and 66.46% better in terms of computational costs. In my opinion, this paper has some major limitations and needs to be modified before being accepted.

(1) The names of variables and functions should be in italics. (2) What is the difference between the contents in Table 2 and Table 3? Why not merge? (3) The experimental section suggests adding some bar charts and graphs to compare the cost of different schemes. (4) The discussion or comparisons with more recent related schemes, such as lightweight trustworthy message exchange in unmanned aerial vehicle networks, icra: an intelligent clustering routing approach for uav ad hoc networks, pbag: a privacy-preserving blockchain-based authentication protocol with global-updated commitment in iovs, instead of conventional schemes are suggested. (5) The writing of the paper needs further improvement.

6. PLOS authors have the option to publish the peer review history of their article (what does this mean?). If published, this will include your full peer review and any attached files.

Reviewer #1: No

Reviewer #2: No

---

## [Author Response · Author response to Decision Letter 0]

31 Oct 2024

A Detailed rebutall letter is attached

---

## [Decision Letter · Decision Letter 1]

12 Nov 2024

A verifiably secure and robust authentication protocol for synergistically-assisted IoD deployment drones

PONE-D-24-39087R1

Dear Dr. Algarni,

We’re pleased to inform you that your manuscript has been judged scientifically suitable for publication and will be formally accepted for publication once it meets all outstanding technical requirements.

Kind regards,

Dr. Md Tarique Jamal Ansari, Ph.D.

Academic Editor

PLOS ONE

Additional Editor Comments (optional):

The authors have successfully addressed all the revision comments. The manuscript is now accepted for publication in its current form.

Reviewers' comments:

Reviewer's Responses to Questions

**Comments to the Author**

1. If the authors have adequately addressed your comments raised in a previous round of review and you feel that this manuscript is now acceptable for publication, you may indicate that here to bypass the “Comments to the Author” section, enter your conflict of interest statement in the “Confidential to Editor” section, and submit your "Accept" recommendation.

Reviewer #1: All comments have been addressed

Reviewer #2: (No Response)

2. Is the manuscript technically sound, and do the data support the conclusions?

Reviewer #1: Yes

Reviewer #2: (No Response)

3. Has the statistical analysis been performed appropriately and rigorously? 

Reviewer #1: Yes

Reviewer #2: (No Response)

4. Have the authors made all data underlying the findings in their manuscript fully available?

Reviewer #1: Yes

Reviewer #2: (No Response)

5. Is the manuscript presented in an intelligible fashion and written in standard English?

Reviewer #1: Yes

Reviewer #2: (No Response)

6. Review Comments to the Author

Reviewer #1: The work have been revised interestingly good. All comments have been addressed in satisfying level.

The recommendation is to accept the work.

Reviewer #2: The author has made some serious revisions to the previous revision suggestions, and I have no new suggestions, so I suggest directly accepting this paper.

7. PLOS authors have the option to publish the peer review history of their article (what does this mean?). If published, this will include your full peer review and any attached files.

Reviewer #1: No

Reviewer #2: No

---

## [Editor Report · Acceptance letter]

19 Nov 2024

PONE-D-24-39087R1 

PLOS ONE

Dear Dr. Algarni, 

I'm pleased to inform you that your manuscript has been deemed suitable for publication in PLOS ONE. Congratulations! Your manuscript is now being handed over to our production team.

Kind regards, 

on behalf of

Dr. Md Tarique Jamal Ansari 

Academic Editor

PLOS ONE